# Tailoring Strictly Proper Scoring Rules for Downstream Tasks: An Application to Causal Inference

Roman Plaud [* 1 2]   Alexandre Perez-Lebel [* 3]   Antoine Saillenfest [2]   Thomas Bonald [1]   Marine Le Morvan [4]
Gaël Varoquaux [4 5]   Matthieu Labeau [1]

## Abstract

Probabilistic models are typically trained using task-agnostic objectives like log-loss, which can lead to significant errors in downstream estimation. This disconnect is especially critical in Inverse Probability Weighting (IPW) for causal inference, where propensity score errors near $0$ and $1$ often lead to high bias and variance. We propose a principled framework for deriving task-specific strictly proper scoring rules by matching the local curvature of the downstream error metric. We apply this to the Average Treatment Effect (ATE) estimation, deriving a closed-form loss and its corresponding canonical probability mapping that can be readily integrated with any model like a neural network or a gradient boosting algorithm. Extensive evaluations on causal inference benchmarks demonstrate that our tailored objective consistently outperforms standard likelihood-based and covariate-balancing approaches.

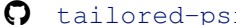 tailored-psr

## 1. Introduction

Many modern applications of machine learning involve a two-step mechanism in which we first estimate conditional probabilities, which then serve as inputs for a specific downstream task. In this setting, which ranges from classification to risk assessment or causal inference, the ultimate target metric is computed on the downstream estimator, not the likelihood of the model itself.

Despite this two-step structure, the training of these probabilistic models often relies on log-loss minimization. While statistically well-principled—minimizing the

Kullback-Leibler divergence to the underlying oracle distribution—this objective is agnostic to the downstream task. This disconnect creates a fundamental mismatch: a model can achieve low log-loss while inducing large errors in the final estimand.

This is particularly striking in Inverse Probability Weighting (IPW, Horvitz & Thompson (1952)) estimators in causal inference, where small estimation errors near the boundaries (*i.e.* approaching $0$ or $1$) of the binary conditional probability distribution can cause the bias and variance of the target estimand to explode (Kang & Schafer, 2007). Standard remedies for this mismatch often rely on *post-hoc* heuristics, such as weight trimming or clipping (Crump et al., 2009), which sacrifice consistency to control variance. Other approaches attempt to reconnect these two steps by internalizing a covariate balance constraint directly within the propensity estimation (Hainmueller, 2012; Imai & Ratkovic, 2014; Zubizarreta, 2015). However, the theoretical link between optimizing covariate balance and minimizing the bias of the target estimand is often tenuous and rarely strictly holds (Bruns-Smith & Feller, 2022). This motivated us to ask:

> *Can we tailor the training objective to the sensitivity of the downstream task?*

In this work, we propose a direct, principled solution: a general framework for deriving task-specific strictly proper scoring rules that internalize this downstream geometry. By establishing an upper bound on the downstream estimator Mean Squared Error (MSE), we analyze its curvature to characterize its *local sensitivity* with respect to probability estimates. We then analytically construct the unique strictly proper scoring rule that minimizes this theoretical bound. This approach bridges the gap between probability learning and downstream estimation, ensuring that the training objective penalizes errors most severely in the regions of the simplex where the downstream task is most sensitive.

While the proposed framework applies more broadly, we illustrate its utility in the estimation of the Average Treatment Effect (ATE) via IPW in causal inference. We derive a novel, strictly proper scoring rule whose associated diver-

[1]Institut Polytechnique de Paris [2]Onepoint, France [3]Fundamental Technologies [4]SODA Team, INRIA Saclay, Palaiseau [5]Probabl, France. Correspondence to: Roman Plaud <plaud.roman@gmail.com>.

*Proceedings of the $43^{rd}$ International Conference on Machine Learning*, Seoul, South Korea. PMLR 306, 2026. Copyright 2026 by the author(s).

gence appropriately penalizes errors near boundaries, where IPW is most vulnerable. Our contributions are as follows:

- We formalize a general framework for constructing tailored strictly proper scoring rules by matching the local curvature of downstream estimation errors.
- We apply this framework to causal inference, deriving a novel, closed-form loss function that minimizes a local upper bound on the Mean Squared Error (MSE) of Inverse Probability Weighting (IPW) estimators.
- We derive the associated canonical probability mapping— the final model activation function required to guarantee convexity with respect to the logits—ensuring stable optimization for modern gradient-based learners.
- We demonstrate via extensive benchmarks that our loss reduces estimation error for IPW, Hajek (Hájek, 1971), and AIPW (Robins et al., 1994) compared to standard likelihood-based or balancing approaches.

**Notation**. To bridge the terminology between causal inference and machine learning, we treat the propensity score $e(x)$ and probability $p(x)$ as interchangeable. Similarly, we formulate our training objectives as strictly proper scoring rules, referred to as loss functions.

## 2. Related Work and Background

### 2.1. Strictly Proper Scoring Rules

We ground our framework in the theory of strictly proper scoring rules, which provides a rigorous basis for probability estimation. A scoring rule $\ell : \{0, 1\} \times [0, 1] \to \mathbb{R}$ is defined as *strictly proper*[1] if the expected score is uniquely minimized when the estimated probability $q$ coincides with the true probability $p$, i.e., $p = \mathrm{argmin}_{q \in [0,1]} \mathbb{E}_{y \sim p}[\ell(y, q)]$.

As established by Buja et al. (2005) and Gneiting & Raftery (2007), every such rule $\ell$ is uniquely characterized (up to constants) by a convex entropy function $H_\ell$, or alternatively its second derivative, the weight function $w_\ell(q) = H_\ell''(q)$. Every proper scoring rule induces a divergence $d_\ell(p, q)$, defined as the difference between the expected score and the negative entropy:

$$d_\ell(p, q) \coloneqq \mathbb{E}_{y \sim p}[\ell(y, q)] - \mathbb{E}_{y \sim p}[\ell(y, p)]. \quad (1)$$

The local *curvature* of this divergence corresponds to the weight function:

$$\left. \frac{\partial^2}{\partial q^2} d_\ell(p, q) \right|_{q=p} = w_\ell(p). \quad (2)$$

For example, the standard log-loss corresponds to the Shannon entropy, inducing the Kullback-Leibler (KL) divergence and the specific weight function $w(q) = \frac{1}{q(1-q)}$.

---

[1]We restrict here to the binary setting as our application falls in this setup, but these definitions can be easily extended to multiclass.

**Canonical Link.** A crucial concept for optimization is the *canonical link function* associated with $\ell$: $g_\ell : [0, 1] \to \mathbb{R}$. It is defined by the relation $g_\ell' = w_\ell$, mapping the probability simplex to logits. In the context of machine learning probabilistic models, this dictates the choice of the final activation function, which must be the inverse canonical link $g_\ell^{-1}$, and which we call in this work the *canonical probability mapping* $\sigma_\ell$. Pairing a loss with this specific activation ensures that the objective function is convex with respect to the linear predictor (the model outputs before the final mapping), guaranteeing stability for gradient-based optimization (Reid & Williamson, 2010). Extending the log-loss example, integrating its weight function yields the logit function $g(q) = \log(\frac{q}{1-q})$ as its canonical link. Consequently, its canonical probability mapping $g^{-1}$ is the standard sigmoid activation function.

### 2.2. Covariate Balancing and Robust Estimation

Standard propensity score estimation typically minimizes the log-loss. While producing consistent probability estimates under correct model specification, this objective ignores the downstream causal task, and a small log-loss can result in catastrophic error on the downstream estimator (Kang & Schafer, 2007).

To mitigate this, prior work attempts to bridge the gap between prediction and estimation through covariate balance constraints—i.e., the degree to which control and treated groups have similar feature distributions after reweighting. Methods like Covariate Balancing Propensity Score (CBPS) (Imai & Ratkovic, 2014) attempt to internalize this goal during probability learning (Shang et al., 2025). Entropy Balancing (Hainmueller, 2012) and Stable Balancing Weights (SBW) (Zubizarreta, 2015) solve directly for weights that satisfy moment conditions, bypassing the probability estimation entirely. Similarly, Kernel Balancing (Hazlett, 2016) employs kernel methods to achieve balance in a reproducing kernel Hilbert space. While balance serves as a proxy for bias reduction (Cannas & Arpino, 2019), this relationship strictly holds only under specific outcome modeling assumptions (Bruns-Smith & Feller, 2022) and often yields asymptotic bias under local model misspecification (Fan et al., 2016).

More recently, methods like RieszNet (Chernozhukov et al., 2022) attempt to learn the Riesz representer of the ATE functional directly. Alternatively, frameworks like Double/Debiased Machine Learning (DML) (Chernozhukov et al., 2018) rely on Neyman orthogonality and cross-fitting to mitigate the asymptotic regularization and overfitting biases inherent in ML nuisance models. However, while DML successfully provides asymptotic guarantees at the estimator level (e.g., via AIPW, Robins et al., 1994), it typically leaves the underlying probability estimation phase untouched.

Closest to our work, Zhao (2019) pioneered the application of proper scoring rule theory to causal inference with Co-variate Balancing Scoring Rules (CBSR), deriving loss functions whose gradients recover specific covariate balancing conditions. However, enforcing these moment conditions necessitates breaking the association between the proper scoring rule and its canonical probability mapping. This results in optimization that is prone to instability (e.g., exploding gradients), preventing the integration of this framework into deep learning architectures.

Our approach contrasts with, and complements, these existing paradigms. Unlike methods that alter the target estimand (e.g., Overlap Weighting; Li et al., 2018) or apply *post-hoc* stabilization like clipping (Crump et al., 2009) and calibration (Deshpande & Kuleshov, 2024; van der Laan et al., 2025; Klaassen et al., 2025), our proposed objective acts as an *ante-hoc* regularizer. Complementary to DML, we internalize the specificity of the downstream task directly into the training process, geometrically aligning the probability estimates with the downstream MSE. Finally, unlike CBSR, we achieve this task-specific tailoring while strictly retaining the canonical probability mapping, guaranteeing stable optimization for modern gradient-based learners.

### 2.3. Background on Causal Inference

In this work, we define the target downstream task as estimating the Average Treatment Effect (ATE) within the potential outcomes framework (Rubin, 1974). We consider a dataset $\mathcal{D} = \{(X_i, T_i, Y_i)\}_{i=1}^N$ with covariates $X_i \in \mathcal{X}$, binary treatment $T_i \in \{0, 1\}$, and observed outcome $Y_i \in \mathbb{R}$.

Under standard assumptions of **consistency**, **unconfoundedness**, and **positivity** (see Appendix E for definitions), the ATE $\tau_{\text{ATE}} = \mathbb{E}[Y(1) - Y(0)]$ is identifiable via the propensity score $e(x) := \mathbb{P}(T = 1 \mid X = x)$:

$$\tau_{\text{ATE}} = \mathbb{E}\left[\frac{Y \cdot T}{e(X)} - \frac{Y \cdot (1-T)}{1 - e(X)}\right]. \quad (3)$$

The standard IPW estimator is constructed by taking the empirical counterpart of Equation 3. Since oracle propensity scores $e(X)$ are unknown, they are replaced by a learned estimate $\hat{e}(X)$ yielding the "plug-in" estimator $\hat{\tau}_{\text{ATE}}$:

$$\hat{\tau}_{\text{ATE}} = \frac{1}{N} \sum_{i=1}^N \left(\frac{Y_i \cdot T_i}{\hat{e}(X_i)} - \frac{Y_i \cdot (1-T_i)}{1 - \hat{e}(X_i)}\right). \quad (4)$$

Our goal is therefore to derive a loss for propensity estimation that minimizes an error related to this specific estimator.

## 3. Tailoring losses to downstream tasks

We propose a methodological framework for tailoring loss functions to downstream estimation tasks. We first present a

general framework for deriving task-specific strictly proper scoring rules (Section 3.1) and then apply it to the specific problem of ATE estimation via Inverse Probability Weighting (Section 3.2).

### 3.1. General Framework: Task-Specific Scoring Rules

Consider a two-step estimation problem. The first step estimates a binary conditional probability[2] $p(x) = \mathbb{P}(T = 1 \mid X = x)$ from sample pairs $(X_i, T_i)$ using a model $\hat{p}$. In the second step, this estimate serves as input for a downstream estimator $\hat{\theta} := \hat{\theta}(\hat{p})$. Since $\hat{p}$ is inevitably imperfect—for example due to finite sample variance or model misspecification—estimation errors in the first step propagate to the second, therefore degrading the final estimation of $\theta$.

Let $\mathcal{E}(\theta, \hat{\theta})$ denote the error metric of the downstream estimator (e.g. MSE). We assume this error can be upper-bounded by the expectation of a divergence $d_{\text{task}}(p, \hat{p})$, an assumption that will later be verified in the framework of causal inference:

> **Assumption 3.1** (Upper-bound of the error metric)**.** There exist constants $C_0 \geq 0$ and $C_1 > 0$, both independent of $\hat{p}$, such that:
>
> $$\mathcal{E}(\theta, \hat{\theta}) \leq C_0 + C_1 \cdot \mathbb{E}_X \left[d_{\text{task}}(p(X), \hat{p}(X))\right]. \quad (5)$$

This formulation isolates the contribution of $\hat{p}$ to the final estimation error. Consequently, minimizing the divergence term on the right-hand side serves as a direct proxy for improving the downstream estimator. We further rely on the following assumption regarding the local geometry of the task error:

> **Assumption 3.2** (Task Sensitivity)**.** For all $p \in (0, 1)$, the task-specific divergence $q \mapsto d_{\text{task}}(p, q)$ is twice continuously differentiable and admits a non-degenerate local minimum at the true value $q = p$. Specifically:
>
> $$\left.\frac{\partial d_{\text{task}}}{\partial q}\right|_{q=p} = 0 \quad \text{and} \quad \left.\frac{\partial^2 d_{\text{task}}}{\partial q^2}\right|_{q=p} > 0. \quad (6)$$

Assumption 3.2 essentially states that the downstream error is minimized at the oracle probability ($q = p$), implying that the first-order derivative vanishes and that the error is locally dominated by a positive second-order term. The Taylor expansion of the task error around $p$ is therefore

---

[2]We denote the binary target variable as $T$ (deviating from the standard machine learning notation $Y$) to avoid ambiguity with the potential outcome variable $Y$ used in the subsequent causal inference application.

given by:

$$d_{\text{task}}(p,q) \underset{q \to p}{=} \frac{1}{2} w_{\text{task}}(p)(p-q)^2 + o\left((p-q)^2\right) \quad (7)$$

where the weight function $w_{\text{task}}(p) := \partial_q^2 d_{\text{task}}(p,q)\big|_{q=p}$ characterizes the *local sensitivity* of the downstream estimator, explicitly identifying which regions of the probability simplex $[0,1]$ contribute most severely to the final error.

### 3.1.1. MATCHING THE LOCAL CURVATURE

Our objective is to train $\hat{p}$ using a loss function $\ell$ that mimics this geometry. Recall that any strictly proper scoring rule is generated by a non-negative weight function $w_\ell$, and its associated divergence defined in Section 2.1 locally behaves as:

$$d_\ell(p,q) \underset{q \to p}{=} \frac{1}{2} w_\ell(p)(p-q)^2 + o((p-q)^2). \quad (8)$$

We therefore derive the optimal task-specific scoring rule by matching the curvature of the loss divergence $d_\ell$ to the curvature of the task divergence $d_{\text{task}}$:

$$w_\ell(p) = w_{\text{task}}(p). \quad (9)$$

We recover the unique proper scoring rule $\ell$ determined by the weight function $w_\ell$ via the following characterization:

> **Proposition 3.3** (Recovery of Tailored Loss). *(Buja et al., 2005).* *Let $w_\ell$ defined by Equation 9. Its corresponding strictly proper scoring rule $\ell(t,q)$ : $\{0,1\} \times [0,1] \to \mathbb{R}$ decomposes as:*
>
> $$\ell(t,q) = t \cdot \ell_1(q) + (1-t) \cdot \ell_0(q), \quad (10)$$
>
> *where the partial losses $\ell_1(\cdot)$ and $\ell_0(\cdot)$ are determined by the differential equations:*
>
> $$\ell_1'(q) = -w_\ell(q)(1-q) \quad and \quad \ell_0'(q) = w_\ell(q)q.$$
>
> *The integration constants are arbitrary and without loss of generality, we set $\ell_1(1) = \ell_0(0) = 0$.*

This tailored loss serves as a convex objective that inherently penalizes errors in proportion to their impact on the downstream task.

### 3.1.2. THEORETICAL GUARANTEE: LINKING LOSS TO DOWNSTREAM ERROR

We now formalize the connection between optimizing our tailored scoring rule and minimizing the downstream error.

> **Proposition 3.4** (Alignment with Downstream Error). *Minimizing the expected risk $\mathcal{R}(\hat{p}) := \mathbb{E}[\ell(T, \hat{p}(X))]$ is equivalent to minimizing the expected tailored divergence $\mathcal{L}(\hat{p}) := \mathbb{E}[d_\ell(p(X), \hat{p}(X))]$. Furthermore, by construction ($w_\ell = w_{\text{task}}$), the tailored divergence matches the task error geometry locally:*
>
> $$d_\ell(p,q) \underset{q \to p}{=} d_{task}(p,q) + o\left((p-q)^2\right). \quad (11)$$

**Proof.** The optimization equivalence follows directly from the aleatoric-epistemic decomposition of strictly proper scoring rules (Gneiting & Raftery, 2007; Kull & Flach, 2015):

$$\mathbb{E}[\ell(T, \hat{p}(X))] = \mathbb{E}[d_\ell(p(X), \hat{p}(X))] + \mathbb{E}[H_\ell(p(X))]$$

Since the entropy $H_\ell(p)$ depends only on the data-generating process and is independent of $\hat{p}$, minimizing $\mathcal{R}$ is identical to minimizing $\mathcal{L}$. The local matching follows from Equation 7 and Equation 8.

This result establishes a direct theoretical link between the training objective and the downstream estimator. While a similar derivation for log-loss would result in minimizing the Kullback-Leibler divergence, our approach minimizes the expected tailored divergence $d_\ell$. Since $d_\ell$ is constructed to match the local geometry of $d_{\text{task}}$ (Eq. 7), minimizing the training loss $\ell$ effectively minimizes the second-order upper bound on the downstream error $\mathcal{E}(\theta, \hat{\theta})$.

More formally, $d_\ell$ acts as the *natural convex surrogate* of the task error. By enforcing $w_\ell = w_{\text{task}}$, we ensure that the loss landscape *locally* penalizes prediction errors exactly in proportion to their impact on the final estimation, while maintaining the *global* convexity of the loss required for stable training.

Finally, to ensure stable gradient dynamics, the derived proper scoring rule $\ell$ must be paired with its canonical probability mapping $\sigma_\ell$ defined by $(\sigma_\ell^{-1})' = w_\ell$. This necessity will be exemplified in Section 3.2.4

Equipped with this general framework for deriving both the loss and its canonical probability mapping, we now proceed to apply it to a concrete challenge: ATE estimation via Inverse Probability Weighting in causal inference.

### 3.2. Application to Causal Inference

To demonstrate the utility of our general framework, we apply it to the estimation of the ATE using the IPW estimator described in Section 2.3.

### 3.2.1. MAPPING ATE ESTIMATION TO THE GENERAL FRAMEWORK

We treat the propensity score $e(x)$ as the conditional probability $p(x)$ from our general framework. To derive the

IPW-specific loss, we map the causal objects directly to the notations introduced in Section 3.1:

- **True Parameter ($\theta$):** The Average Treatment Effect:

$$\theta := \tau_{\text{ATE}} = \mathbb{E}[Y(1) - Y(0)]$$

- **Downstream Estimator ($\hat{\theta}$):** The Inverse Probability Weighting estimator:

$$\hat{\theta} := \hat{\tau}_{\text{ATE}} = \frac{1}{N} \sum_{i=1}^{N} \left( \frac{Y_i T_i}{\hat{e}(X_i)} - \frac{Y_i (1 - T_i)}{1 - \hat{e}(X_i)} \right)$$

- **Error Metric ($\mathcal{E}$):** The Mean Squared Error (MSE) of the estimator:

$$\mathcal{E}(\theta, \hat{\theta}) := \mathbb{E}\left[ (\hat{\theta} - \theta)^2 \right]$$
$$= \underbrace{(\mathbb{E}[\hat{\theta}] - \theta)^2}_{\text{Bias}(\hat{\theta})^2} + \underbrace{\mathbb{E}[(\hat{\theta} - \mathbb{E}[\hat{\theta}])^2]}_{\text{Var}(\hat{\theta})}$$

Our goal is to find the specific divergence $d_{\text{task}}$ that satisfies Assumption 3.1 to get an upper bound on this error metric.

### 3.2.2. DERIVING THE TASK-SPECIFIC DIVERGENCE

By analyzing the bias and variance components separately, we derive a tractable upper bound on the total downstream error:

---

**Theorem 3.5** (Upper Bound on Downstream MSE).
*Assume the following conditions hold:*

- ***Causal Identifiability:** Consistency, unconfoundedness, and strict positivity hold.*
- ***Bounded Variance:** $\mathbb{E}[Y(t)^2 \mid X] \leq M^2$ almost surely for $t \in \{0, 1\}$ and some $M < \infty$.*
- ***Cross-Fitting:** Propensity scores $\hat{e}(X)$ are estimated via cross-fitting, allowing $\hat{e}$ to be treated as a fixed, out-of-sample function.*

*Then, the MSE of the plug-in IPW estimator is upper-bounded as:*

$$\mathcal{E}(\theta, \hat{\theta}) \leq C_0 + C_1 \cdot \mathbb{E}\left[d_{task}(e(X), \hat{e}(X))\right] \quad (12)$$

*where $C_0, C_1$ are constants independent of $\hat{e}$. The task divergence $d_{task}(p, q) = d_{bias}(p, q) + d_{var}(p, q)$ is defined by:*

$$d_{bias}(p, q) = \left(\frac{p}{q} - 1\right)^2 + \left(\frac{1-p}{1-q} - 1\right)^2$$
$$d_{var}(p, q) = p\left(\frac{1}{q} - \frac{1}{p}\right)^2 + (1 - p)\left(\frac{1}{1-q} - \frac{1}{1-p}\right)^2$$

---

**Proof Sketch.** The proof relies on the bias-variance decom-

position of the MSE.

*(i) Bias Term:* The bias of the treated term is $\mathbb{E}[\mu_1(X)(\frac{e}{\hat{e}} - 1)]$, where $\mu_1(X) := \mathbb{E}[Y(1) \mid X]$ denotes the conditional expected potential outcome. Applying the Cauchy-Schwarz inequality allows us to separate the outcome magnitude from the propensity error, yielding an upper bound proportional to $\mathbb{E}[(\frac{e}{\hat{e}} - 1)^2]$, which constitutes $d_{\text{bias}}$.

*(ii) Variance Term:* The variance of the IPW estimator scales with the expected squared error of the inverse-weighted outcomes. Specifically, the error contribution from the treated term behaves as $\mathbb{E}[Y(1)^2 e(\frac{1}{\hat{e}} - \frac{1}{e})^2]$. Bounding the conditional outcome moments by $M^2$ allows us to extract the term $d_{\text{var}}$. The constants $C_0$ and $C_1$ absorb the irreducible oracle variance and the outcome magnitude bounds. (See Appendix E for the complete proof).

The bias component $d_{\text{bias}}$ explodes quadratically ($1/q^2$) as $q \to 0$, while the variance component $d_{\text{var}}$ introduces a cubic penalty ($1/q^3$), reflecting both the extreme sensitivity of IPW bias and variance to small propensity scores.

*Remark* 3.6 (Extension to Robust Estimators). While Theorem 3.5 is, for clarity, restricted to the standard IPW estimator, we show in Appendix E.3.1 that the MSE upper bounds for the Hajek (Hájek, 1971) and Augmented Inverse Probability Weighting (AIPW) (Robins et al., 1994) estimators share the same mathematical structure. Specifically, the MSE error is driven by the exact same task-specific divergence $d_{\text{task}}$ (differing only by scaling constants). Consequently, the tailored loss derived here remains identical for these robust estimators.

### 3.2.3. MATCHING LOCAL CURVATURE

As explained in Section 3.1, we approximate $d_{\text{task}}$ locally using the curvature matching framework. The total task curvature $w_{\text{task}}(p)$ is derived by summing the second derivatives of the bias and variance terms which yields:

$$w_{\text{task}}(p) = \underbrace{\left(\frac{2}{p^2} + \frac{2}{(1-p)^2}\right)}_{\text{Curvature from } d_{\text{bias}}} + \underbrace{\left(\frac{2}{p^3} + \frac{2}{(1-p)^3}\right)}_{\text{Curvature from } d_{\text{var}}} \quad (13)$$

Setting $w_\ell = w_{\text{task}}$ and integrating this weight function via Prop 3.3 (See Appendix F) yields our tailored objective.

---

**Definition 3.7** (IPW-Tailored Proper Scoring Rule).
The strictly proper scoring rule $\ell$ that locally minimizes the upper bound on the IPW MSE is defined as:

$$\ell(t, q) = t \left[ \frac{1}{q^2} - \frac{2}{1-q} + 2\log\left(q(1-q)\right) \right]$$
$$+ (1-t) \left[ \frac{1}{(1-q)^2} - \frac{2}{q} + 2\log\left(q(1-q)\right) \right]$$

---

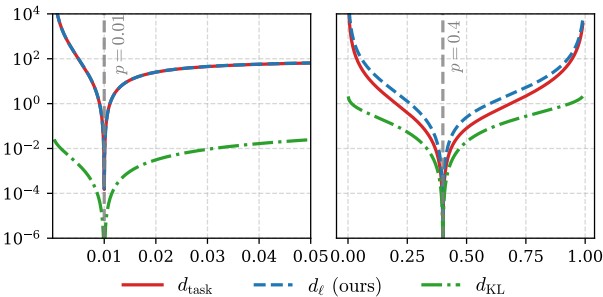

*Figure 1.* $d_\ell$ **locally behaves as** $d_{\text{task}}$. Log-scale plots of $q \mapsto d(p, q)$ for $d \in \{d_{\text{task}}, d_\ell, d_{\text{KL}}\}$. The vertical dashed lines indicate the true probability $p$, with $p = 0.01$ in the left panel and $p = 0.4$ in the right panel.

Figure 1 illustrates the tight alignment between the induced divergence $d_\ell$ and the theoretical bound $d_{\text{task}}$. The match is blatant near the boundaries, where the standard KL divergence largely underestimates the error magnitude.

### 3.2.4. DERIVING THE IPW CANONICAL LINK

The derivation of the loss $\ell$ is only half the solution; training a machine learning model with such a loss necessitates careful design of the final layer activation function.

As derived in Eq. 13, the task-specific weight $w_\ell(p)$ scales with $1/p^3$ and $1/(1-p)^3$ when approaching 0 and 1. If we were to naively pair this loss with a standard Sigmoid activation $\sigma$, the gradient with respect to the logit $z$ would be:

$$\frac{\partial \ell}{\partial z} = w_\ell(p)(p - y) \cdot \underbrace{\sigma'(z)}_{=p(1-p)} . \tag{14}$$

This leads to numerical instability because as predictions approach 0, the magnitude of the weight function ($\approx 1/p^3$) dominates the magnitude of the sigmoid derivative ($\approx p$). This results in gradients that scale as $1/p^2$. A similar reasoning applies when $p \to 1$, leading us to conclude that gradients explode precisely in the regions where accurate estimation is most critical. This is empirically verified in Appendix B.1.

To resolve this, we employ the strategy established in Section 3.1.2: we pair the tailored scoring rule $\ell$ with its corresponding canonical probability mapping $\sigma_\ell$. Defined by the differential equation $(\sigma_\ell^{-1})'(p) = w_\ell(p)$, this activation ensures that the mapping derivative exactly cancels the weight function, reducing the gradient to a stable linear residual $\frac{\partial \ell}{\partial z} = p - y$, which cannot explode or vanish. In our case, solving this differential equation (and setting the integration constant such that $\sigma_\ell(0) = 0.5$) corresponds for $z \in \mathbb{R}$, finding $p$ such that:

$$z = \int \left( \frac{2}{p^2} + \frac{2}{(1-p)^2} + \frac{2}{p^3} + \frac{2}{(1-p)^3} \right) dp. \tag{15}$$

Integration is immediate but inverting Equation 15 requires solving a polynomial equation, which, in our specific case, reduces to finding the largest real root of the following depressed quartic equation (See Appendix F.3 for full derivation) for an auxiliary variable $u = \frac{1}{p(1-p)}$:

$$u^4 - 12u^2 - 16u - z^2 = 0. \tag{16}$$

This consequently yields the following mapping:

**Definition 3.8** (Canonical probability mapping). Let $z \in \mathbb{R}$ and $u_0(z)$ be the largest root of Equation 16. The canonical link associated with $\ell$ is defined as:

$$\sigma_\ell(z) = \frac{1}{2} \left( 1 + \text{sign}(z) \sqrt{1 - \frac{4}{u_0(z)}} \right) \tag{17}$$

*Remark* 3.9 (Deep Learning Implementation). While the forward pass requires evaluating the closed-form root of a quartic equation, the backward pass can be optimized. Because $\ell$ and $\sigma_\ell$ form a canonical pair, the gradient w.r.t. the logits simplifies to $p - y$. For deep learning models, practitioners can either rely on standard automatic differentiation or implement a custom backward pass that directly applies this simple gradient, bypassing the need to differentiate through the root-finding routine. We detail these implementation choices in Appendix C.2 and computation trade-off in Appendix B.4.

*Remark* 3.10 (Bias-Variance Trade-off in Link Functions). We note that excluding the variance terms ($1/p^3$) from the weight function simplifies the inversion to a quadratic equation, yielding a lighter "Bias-Only" loss. While computationally simpler, our experiments suggest full MSE loss provides superior results (See Appendix F.4 for details).

Equipped with our tailored proper scoring rule and its canonical probability mapping, we now evaluate the framework against standard propensity estimators on established causal inference benchmarks.

## 4. Experiments

A fundamental challenge in validating causal inference methodologies is the absence of ground truth counterfactuals in real-world data. To rigorously assess the effectiveness of our proposed tailored scoring rule, we follow the standard benchmarking protocol in the literature by relying on semi-synthetic datasets which combine real-world covariates with simulated outcomes. This preserves realistic feature correlations while ensuring that the true ATE is known.

Our experimental evaluation is structured in two parts: first, we compare our method against specialized balancing or post-hoc modification baselines on low-dimensional benchmarks using linear backbones. Second, we evaluate the scal-

ability and effectiveness of our loss as a drop-in replacement for log-loss in higher-dimensional settings (ACIC 2017).

We emphasize here that our loss-mapping combination is inherently architecture-agnostic, acting as a drop-in replacement for log-loss in any gradient-based learner.

### 4.1. Standard Benchmarks: Comparisons with Balancing Methods

**Datasets.** We evaluate our method on three semi-synthetic benchmarks: **IHDP** (Hill, 2011b), **Jobs** (LaLonde, 1986; Shalit et al., 2017) and **Kang & Schafer** (Kang & Schafer, 2007).

**Baselines.** We compare our Tailored IPW Loss $\ell$ paired with its canonical probability mapping $\sigma_\ell$ against three distinct categories of propensity score estimators:

1. **Likelihood-based:** Standard Logistic Regression trained with log-loss (MLE).
2. **Heuristics:** MLE refined by post-hoc **Trimming** or **Clipping**.
3. **Balancing Methods:** Estimators that directly target sample covariate balance, including **CBPS** (Imai & Ratkovic, 2014), **CBSR** (Zhao, 2019), Entropy Balancing (Hainmueller, 2012), and **SBW** (Zubizarreta, 2015).

Details about baselines implementations can be found in Appendix C.3.

**Protocol.** For these low-dimensional benchmarks, we restrict all propensity estimators to a linear specification. We evaluate downstream performance using **IPW**, **Hajek** (Hájek, 1971), and **AIPW** (Robins et al., 1994) estimators, implementing 10-fold cross-fitting (see Appendix C.2 for details). Each benchmark consists of multiple independent simulation runs (ranging from 10 to 2000), providing distinct realizations of treatment assignments and outcomes.

**Metrics.** Our primary evaluation metric is the Root Mean Squared Error (RMSE) of the ATE, computed across all simulation runs. As the square root of our targeted MSE, this metric directly reflects the objective optimized by our tailored scoring rule. In accordance with standard practices, we also report the Mean Absolute Error (MAE) and the average bias to provide a comprehensive assessment of estimator performance.

**Results.** Figure 2 summarizes the macro-average distribution of standardized errors (standardized vs. the median estimator) errors across all simulation runs (exact methodology is described in Appendix C). The visualization confirms that tailoring the scoring rule to the downstream task yields consistent performance gains.

- **MAE & RMSE:** While likelihood-based methods (Logistic) and balancing approaches (CBPS, CBSR) can be

inconsistent, our tailored loss $\ell$ demonstrates superior performance, especially for the vanilla IPW estimator. It achieves the lowest standardized MAE and RMSE by shifting the error distribution to the left.
- **Bias Control:** In the Bias panel, while our method appears to be slightly biased downwards, the average relative bias of our method is the closest to the Target (0.0).
- **Rankings:** The Mean Rank panel provides a holistic view of results. Our method secures the lowest (best) rank across all three downstream estimators. While post-hoc heuristics like Trimming can be competitive in specific MAE or RMSE realizations, the ranking confirms that our approach is the most consistent top-performer across all the datasets.

Notably, the gains are largest for the vanilla IPW estimator, which lacks the inherent normalization of Hajek estimator or the protection of the doubly-robust AIPW. This confirms our central hypothesis: by internalizing the geometric sensitivity of the IPW objective, our loss function prioritizes accurate estimation exactly where the estimator is most vulnerable *i.e.* at the boundaries of the simplex.

Next, we shift toward more recent and complex dataset : ACIC data challenges.

### 4.2. ACIC 2017

To evaluate performance in higher-dimensional regimes, we rely on the ACIC 2017 Data Challenge (Hahn et al., 2019). This benchmark combines real-world covariates ($N = 4802$ observations, $d = 58$ dimensions) with 32 distinct data-generating processes (DGPs). These DGPs are specifically designed to stress-test estimators against varying degrees of non-linearity and, crucially, varying selection strength—*i.e.*, the extent to which true propensity scores are concentrated near the boundaries (0 and 1), where estimation errors have the highest impact on downstream ATE accuracy. **Setup.** Rather than an extensive benchmark against specialized state-of-the-art architectures (Künzel et al., 2019; Wager & Athey, 2018; Hahn et al., 2020), we assess the practical utility of our framework as a drop-in replacement for the standard log-loss within two model architectures: a Multi-Layer Perceptron (MLP) and Gradient Boosting (XGBoost (Chen & Guestrin, 2016)). We selected these models to cover the two dominant paradigms in tabular learning (neural vs. tree-based). This design highlights a key practical advantage: while adapting covariate balancing objectives to complex backbones is non-trivial, our method requires only a simple loss replacement. For each architecture, we therefore isolate the impact of the loss function by comparing the downstream ATE estimation error under two optimization regimes:

1. **Standard Training:** Minimizing the log-loss
2. **Tailored Training:** Minimizing the proposed MSE-

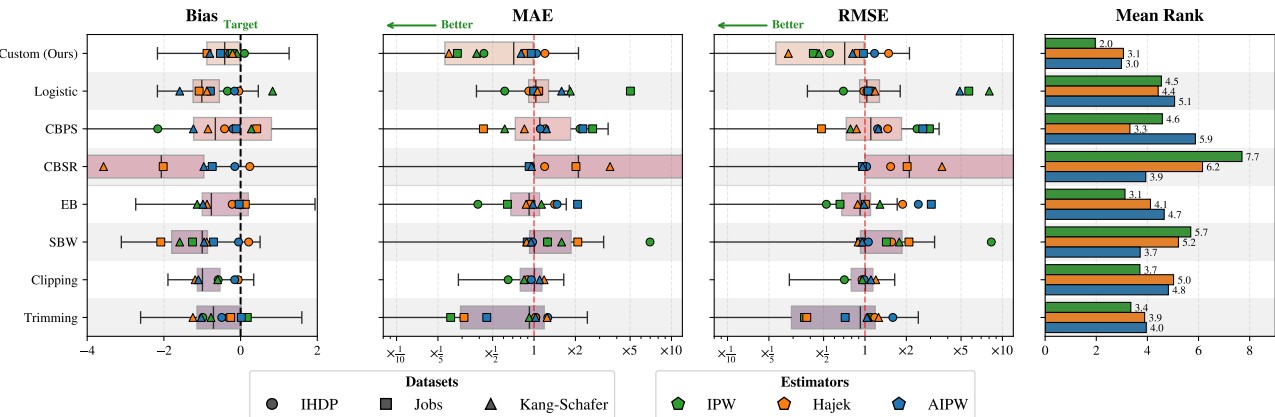

*Figure 2.* **Comparative performance analysis of treatment effect estimation methods across multiple benchmarks**. All metrics are standardized by dividing the error of each method by the median absolute error of all methods within a specific simulation and estimator configuration. The left panels illustrate the distribution of Normalized Bias, MAE, and RMSE across all simulation runs; the objective is to center results on the black dotted line for Bias (0.0) and to achieve the lowest possible values for MAE and RMSE. Logarithmic scales for MAE and RMSE use a red dashed line at 1.0 to denote median baseline performance, with x-axis ticks indicating multiplicative error factors. Overlaid scatter points distinguish performance across datasets (IHDP: ○, Jobs: □, Kang-Schafer: △) and estimators (IPW: green, Hajek: orange, AIPW: blue). The rightmost panel shows the mean rank of each method, calculated for each simulation and averaged across a given dataset and for a given estimator, where lower values indicate more consistent superior performance across diverse dataset.

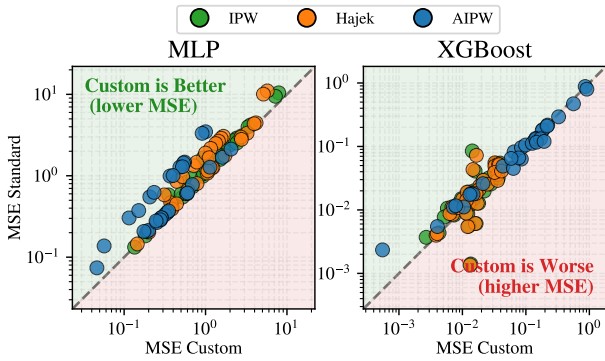

*Figure 3.* **Pairwise RMSE Comparison (ACIC 2017).** Each point represents the average RMSE for a specific estimator-setup pair (32 DGPs × 3 estimators). Points in the upper-left (green) region indicate configurations where the proposed tailored loss yields lower RMSE than the standard log-loss baseline. The tailored objective consistently outperforms the baseline across IPW, Hajek, and AIPW estimators.

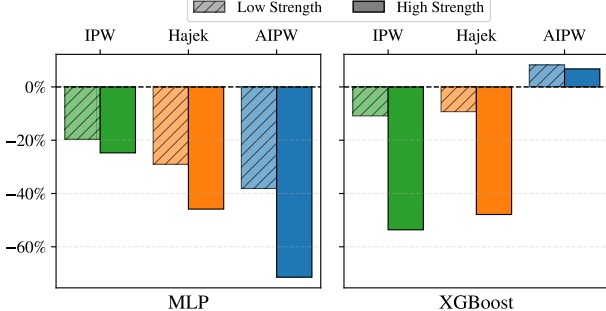

*Figure 4.* **Relative RMSE Improvement by Regime.** Relative RMSE improvement of the tailored loss compared to the log-loss baseline, stratified by selection strength intensity. The decrease in RMSE is most pronounced in high selection strength regimes, validating the theoretical motivation of tail-sensitive scoring rules.

aware proper scoring rule coupled with its canonical probability mapping.

**Global Performance.** We first quantify the benefit of replacing the standard log-loss with our tailored objective. Figure 3 presents a pairwise comparison of the RMSE for each combination of setup and downstream estimator (96 pairs in total). Points situated above the diagonal $y = x$ indicate scenarios where our tailored loss reduces the estimation error compared to the baseline. The results demonstrate a consistent advantage: for the MLP backbone, our method improves performance in 95 out of 96 configura-

tions. Similarly, for Gradient Boosting, our method yields superior estimates in 62 out of 96 cases. **Impact of Selection Strength.** To elucidate the source of these improvements, we stratify the results based on the intensity of the selection strength. Half of the 32 DGPs are characterized by strong selection (Hahn et al., 2019), defined here as scenarios where true propensity scores are more concentrated near the boundaries (0 and 1). As formalized in Section 3, standard IPW estimators are most vulnerable in these regimes, as small predictive errors for these extreme probabilities can lead to massive bias and variance inflation.

Figure 4 displays the relative RMSE improvement achieved by our method. We observe that the performance gains are higher in high-strength settings. While our tailored

loss consistently outperforms the log-loss in low-selection regimes (except for XGBoost with AIPW), the gap widens when selection is strong. This confirms that our objective provides the most benefit in scenarios where downstream estimators are typically most vulnerable.

## 5. Conclusion

In this work, we addressed the fundamental mismatch between standard probabilistic training objectives and downstream estimation tasks. By formalizing a framework that aligns the local curvature of strictly proper scoring rules with the geometry of the downstream error, we derived a tailored loss that minimizes a local upper bound on the estimation error. Applied to the context of causal inference, our proposed IPW-tailored loss effectively penalizes the near-boundaries probabilities errors that drive bias and variance in ATE estimation.

A strength of our approach is its modularity. Unlike methods that require specialized architectures or complex optimization routines, our framework provides a *plug-and-play* loss and probability mapping pair. This acts as a drop-in replacement for the standard log-loss, allowing practitioners to leverage the tailored objective across any model.

While we focused on ATE estimation via IPW, future work could include applying this framework to other causal estimands, such as the Average Treatment Effect on the Treated (ATT), the Conditional Average Treatment Effect (CATE), or to other robust estimators like Targeted Maximum Likelihood Estimation (TMLE) (van der Laan & Rose, 2011). We believe this work serves as a proof of concept and we hope these results encourage the community to adopt these tailored objectives that explicitly account for the downstream estimation task.

## Limitations

While our framework demonstrates empirical and theoretical results, we acknowledge several limitations.

**Local Curvature Matching**. Our derivation relies on a local Taylor expansion of the task-specific error. While this provides a principled framework for aligning the loss geometry with the downstream objective near the true probability $p$, it does not offer formal global guarantees. Specifically, we do not have a theoretical guarantee that the loss remains the tightest possible bound in regions far from $p$. Nonetheless, our empirical results suggest that matching local curvature is a sufficient and effective heuristic to guide the model toward significantly more robust estimations in practice.

**Computational Trade-offs.** Replacing a standard, highly optimized objective like log-loss with a custom loss introduces a computational overhead. During the forward pass,

our method requires solving a quartic equation, which is more expensive than the standard Sigmoid activation. While we mitigate this in deep learning architectures by using a simplified analytical backward pass $(p-y)$, the total training time remains slightly higher than standard likelihood-based models. We provide a detailed quantification of these timing trade-offs across different backbones in Appendix B.4.

**Tractability of the Canonical Mapping.** A core strength of our approach is the derivation of the canonical probability mapping $\sigma_\ell$ to ensure stable optimization. However, a closed-form solution for this mapping is not guaranteed for every arbitrary task. For the IPW-ATE objective, this required solving a fourth-degree polynomial; for other downstream tasks, the resulting differential equations may not yield analytical inverses, potentially requiring practitioners to rely on numerical root-finding or approximation methods.

## Impact Statement

This paper presents work whose goal is to advance the field of machine learning. There are many potential societal consequences of our work, none of which we feel must be specifically highlighted here.

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

# Appendix

# A. Supplementary Related Work

In this section, we supplement the discussion of prior works related to our paper.

## A.1. Downstream Task Adaptation

Traditionally, predictive models maximize task-agnostic likelihood (e.g., log-loss), decoupling prediction from its final utility. However, optimal prediction often misaligns with downstream objectives due to differing bias-variance trade-offs (Fernández-Loría & Provost, 2022). This gap is bridged through downstream task adaptation. In classification, this includes *post-hoc* cost-sensitive decoding (Elkan, 2001; Perez-Lebel et al., 2025; Plaud et al., 2024; 2025) or *ante-hoc* structural modifications to the training objective (Osokin et al., 2017). The latter encompasses consistent surrogate bounds for non-decomposable metrics (e.g., F1, AUC) (Dembczynski et al., 2011; Koyejo et al., 2014; Eban et al., 2016), and continuous relaxations like Fenchel-Young losses that embed structured geometries via generalized entropy (Blondel et al., 2020).

Beyond classification, the "predict-then-optimize" framework (Bertsimas & Kallus, 2020) spawned Decision-Focused Learning (DFL) (Donti et al., 2017; Mandi et al., 2024) and Smart Predict-then-Optimize (Elmachtoub & Grigas, 2022), which aligns predictive and decision losses by backpropagating through differentiable solvers (Amos & Kolter, 2017). While sharing DFL's conceptual goal, our causal framework operates without complex solvers or discrete decoders. Because Inverse Probability Weighting (IPW) is a closed-form plug-in formula, we achieve task-awareness directly by analytically bounding its Mean Squared Error (MSE) and embedding this continuous geometry into the upstream objective.

## A.2. Robust Causal Estimation Paradigms

Beyond direct propensity modification, downstream causal robustness is traditionally achieved through doubly robust estimators like Augmented IPW (Robins et al., 1994) or Targeted Maximum Likelihood Estimation (TMLE) (van der Laan & Rose, 2011), alongside Double/Debiased Machine Learning (DML) (Chernozhukov et al., 2018) to eliminate nuisance biases. Concurrently, deep causal architectures focus on learning regularized outcome representations (Shalit et al., 2017; Hill, 2011a). Recent advancements push this paradigm further via in-context learning; frameworks like Do-PFN (Robertson et al., 2026) leverage transformers pre-trained on massive synthetic causal datasets to perform zero-shot causal estimation on tabular data. Bridging representation learning and estimation, Dragonnet (Shi et al., 2019) appends an asymptotic TMLE regularization term to standard predictive losses. Unlike methods relying on *post-hoc* clipping (Crump et al., 2009), additive regularization, or zero-shot inference, our tailored proper scoring rule structurally redesigns the primary classification loss itself. Serving as a drop-in replacement for log-loss, it remains fundamentally complementary to advanced downstream frameworks like DML or TMLE.

# B. Extended Analysis and Ablations

## B.1. The Necessity of the canonical probability mapping

In Section 3.2.4, we argued that pairing the tailored scoring rule with its canonical probability mapping is theoretically required for optimization stability. To validate this empirically, we trained an (unregularized) Logistic regression with stochastic gradient descent on a simulation of the Kang-Schafer dataset using two configurations:

1. **Mismatched:** The tailored loss $\ell$ paired with a standard Sigmoid activation $\sigma$.
2. **Matched (Ours):** The tailored loss paired with the derived quartic canonical link $g_\ell$.

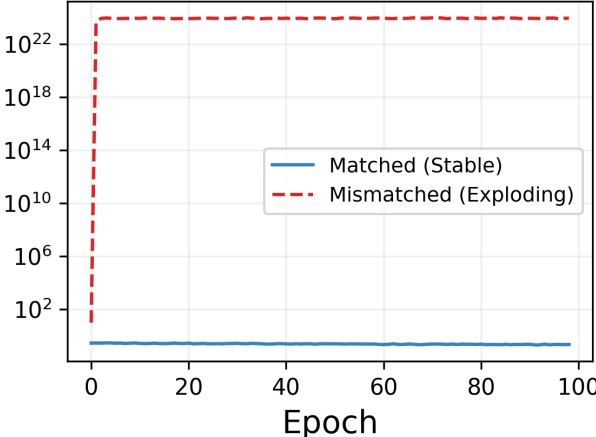

*Figure B.1.* **Optimization Stability Analysis.** Comparison of gradient norms during training. The **Mismatched** configuration (red) suffers from immediate catastrophic divergence, reaching norms of $10^{24}$. In contrast, the **Matched** canonical link (blue) maintains stable gradients ($< 0.28$) throughout training.

Figure B.1 demonstrates the critical difference in stability. The **Mismatched** configuration suffers from catastrophic divergence almost immediately after initialization. The gradient norms explode to $10^{24}$ within the first few iterations, driven by the weight term $w_\ell(p) \cdot \sigma'(z)$, which scales as $O(1/p^2)$ near the boundaries. Conversely, the **Matched** configuration remains numerically stable throughout the entire training process, with gradient norms never exceeding $0.275$. This confirms that the canonical link successfully cancels the curvature divergence, ensuring bounded gradients even in the presence of extreme predictions.

## B.2. Ablation Study: Validating the Variance Penalty

In Remark 3.10, we highlighted a structural trade-off between two tailored scoring rules derived via curvature matching:

1. **Bias-Only Loss:** Derived by matching only the bias curvature ($1/p^2$). It benefits from a simpler quadratic canonical link but ignores the variance of the IPW weights.
2. **Full MSE Loss (Ours):** Derived by matching the complete MSE bound. It requires a quartic canonical link but includes an additional $O(1/p^3)$ penalty term specifically targeting weight variance.

Here, we empirically investigate whether the additional complexity of the Full MSE loss is justified. We compare both objectives across three benchmarks (IHDP, Jobs, Kang-Schafer) using three distinct estimators (IPW, Hajek, AIPW).

Table B.1 demonstrates that while all methods maintain comparable performance on standard benchmarks (IHDP and Jobs), the choice of loss becomes critical in difficult regimes. In the Kang-Schafer simulation—characterized by high lack of overlap—the Log-Loss suffers from catastrophic divergence (RMSE $\approx 66.9$). The Full MSE loss perform better than the Bias-only loss.

*Table B.1.* **Impact of Variance Control.** RMSE of the ATE across three benchmarks (lower is better). While the standard Log-Loss and Bias-Only loss perform comparably in standard settings (IHDP, Jobs), the **Full MSE Loss** (Ours) provides better results in the Kang-Schafer environment.

| OBJECTIVE | IHDP | | | JOBS | | | KANG & SCHAFER | | |
| --- | --- | --- | --- | --- | --- | --- | --- | --- | --- |
| | IPW | HAJEK | AIPW | IPW | HAJEK | AIPW | IPW | HAJEK | AIPW |
| LOG-LOSS (BASELINE) | 0.51 | **0.16** | **0.19** | 0.46 | 0.08 | 0.09 | 66.90 | 8.12 | 46.51 |
| BIAS-ONLY LOSS | **0.27** | 0.25 | 0.21 | 0.04 | **0.05** | 0.08 | 3.39 | 2.62 | 5.77 |
| **FULL MSE (OURS)** | 0.34 | 0.29 | 0.22 | **0.03** | 0.07 | **0.08** | **2.68** | **1.95** | **5.07** |

## B.3. Visualizing the Canonical Link

To understand how the proposed method achieves robustness, we visualize the canonical link function $g_\ell(z)$ as well as the mapping associated with the Bias-only loss and compared them to the standard Sigmoid.

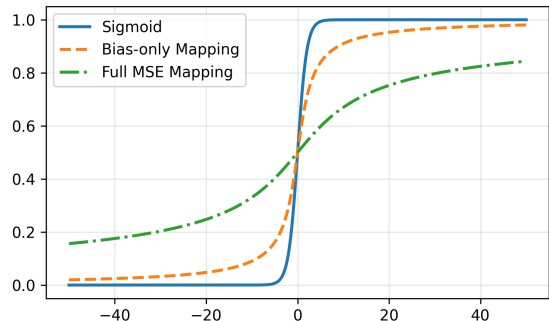

*Figure B.2.* **Comparison of Canonical Link Functions.** We visualize the mapping $z = g(p)$ for the standard Logit (blue), the Bias-Only quadratic link (orange), and the proposed Full MSE quartic link (green). Note that the Full MSE link grows significantly faster near the boundaries ($p \to 0, 1$).

As illustrated in Figure B.2, our canonical link function approaches the asymptotes (0 and 1) much more slowly than the Sigmoid function. For a large logit input $z = 10$, Sigmoid yields $p \approx 0.99995$, whereas our link yields $p \approx 0.7$. This "stretching" of the probability space prevents the model from assigning extreme probabilities.

## B.4. Computational Overhead

As discussed in Section 5, replacing a highly optimized standard objective like the log-loss (Binary Cross-Entropy) with our tailored proper scoring rule introduces a computational overhead. Specifically, our canonical probability mapping requires computing the roots of a quartic equation, which is mathematically more expensive than the standard Sigmoid function.

To precisely quantify this computational trade-off and evaluate the efficacy of our mitigation strategies, we designed a targeted training throughput benchmark.

**Experimental Setup.** We evaluate the models on a large-scale synthetic classification dataset ($N = 10^6$ samples, $d = 100$ features) to ensure that the timing differences are statistically significant and reflect the algorithmic behavior rather than system overhead. All models are trained for a fixed number of iterations without validation-based early stopping.

**Configurations Evaluated.** For the Multi-Layer Perceptron (MLP) backbone (3 hidden layers of 100 neurons), we compare:

1. **Baseline MLP:** Standard Log-Loss paired with a Sigmoid activation.
2. **Naive Custom MLP:** Our tailored loss and quartic mapping, using standard automatic differentiation (Autograd).
3. **Optimized Custom MLP (Ours):** Our tailored loss utilizing the explicit custom backward pass ($\frac{\partial \ell}{\partial z} = p - y$), bypassing the need to backpropagate through the quartic solver.

For the Gradient Boosting (XGBoost) backbone, we compare three configurations to isolate mathematical complexity from

software overhead:

1. **Native Baseline:** The highly optimized `binary:logistic` objective.
2. **Python Logistic:** A custom objective executing standard logistic gradients via a Python callback, serving as a control for the software 'callback tax'.
3. **Custom Quartic (Ours):** Our tailored loss evaluating the quartic canonical inverse to supply the analytical gradient ($g$) and Hessian ($h$).

**Results and Discussion.** The results of the benchmark are summarized in Table B.2.

*Table B.2.* **Propsensity score Training Throughput Benchmark.** Relative training time across configurations on a dataset of $N = 10^6, d = 100$. XGBoost control rows isolate mathematical overhead from Python callback latency.

| BACKBONE | OBJECTIVE MATH | EVALUATION ENGINE | TOTAL TIME | RELATIVE TIME |
|---|---|---|---|---|
| MLP (BASELINE) | LOG-LOSS (SIGMOID) | PYTORCH NATIVE | 163.26s | 1.00× |
| MLP (NAIVE CUSTOM) | QUARTIC ROOT | AUTOGRAD | 185.40s | 1.14× |
| MLP (OPTIMIZED CUSTOM) | QUARTIC ROOT | ANALYTICAL ($p-y$) | 166.79s | **1.02×** |
| XGBOOST (NATIVE BASELINE) | LOG-LOSS (SIGMOID) | NATIVE C++ | 7.25s | 1.00× |
| XGBOOST (PYTHON LOGISTIC) | LOG-LOSS (SIGMOID) | PYTHON CALLBACK | 8.16s | 1.12× |
| XGBOOST (CUSTOM QUARTIC) | QUARTIC ROOT | PYTHON CALLBACK | 36.50s | 5.03× |

The benchmark reveals three key insights regarding the practical implementation of our framework:

- **The Autograd Overhead:** The *Naive Custom MLP* incurs a noticeable performance penalty (a $14\%$ increase in training time). This confirms that while our loss is fully differentiable, forcing the autograd engine to step through the quartic solver is inefficient.
- **Canonical Link Efficiency in PyTorch:** By implementing the analytical backward pass, the *Optimized Custom MLP* recovers almost all of this performance. Because the model computes the probabilities $p$ efficiently during the forward pass and simply reuses them for the $p - y$ gradient, the remaining mathematical overhead is small (approximately $2\%$).
- **Tree-Based Mathematical Overhead:** Unlike PyTorch, XGBoost's custom objective API receives raw logits $z$ and must compute the probabilities $p$ from scratch to evaluate both the gradient ($g = p - y$) and the Hessian ($h = 1/w_\ell(p)$) at every boosting step. As demonstrated by the *Python Logistic* control, the software "callback tax" accounts for only a minor $\sim 12\%$ slowdown. The jump to $36.50$ seconds is fundamentally a mathematical penalty caused by evaluating the quartic mapping on CPU millions of times. While this results in a $\sim 4.5\times$ slowdown relative to the Python baseline, processing over 2 million samples per second remains extremely tractable for practical tabular datasets.

These results confirm that the canonical pair allows deep learning implementations to remain as fast as standard likelihood methods, while tree-based adaptations incur a manageable cost.

# C. Detailed Experimental Setup

## C.1. Datasets

**IHDP.** The Infant Health and Development Program (IHDP) benchmark is derived from a randomized experiment studying the effect of home visits on cognitive test scores. We use the standard semi-synthetic version provided by Hill (2011b), which consists of 747 units (139 treated, 608 control) and 25 covariates describing the children and their mothers.

**Jobs.** Adapted from the LaLonde dataset (LaLonde, 1986) by Shalit et al. (2017). It focuses on evaluating the effect of job training on employment status. The covariates are real, while the outcome $Y$ is simulated to provide a ground truth for evaluation.

**Kang & Schafer.** Originally introduced as a challenge for missing data methods (Kang & Schafer, 2007), this benchmark is a standard "stress test" for causal estimators due to its severe model misspecification and weak overlap. We adapt it to the ATE setting by treating the missingness indicator as a treatment assignment $T$ and setting the Average Treatment Effect to 10. This simulation study is specifically designed to assess estimator robustness under model misspecification and weak overlap. For $N = 2000$ units, latent covariates $Z_i \sim \mathcal{N}(0, I_4)$ are generated. The true propensity score and outcome are linear and logistic functions of these latent terms:

$$e(z) = \sigma(-z_1 + 0.5z_2 - 0.25z_3 - 0.1z_4)$$

$$Y = 210 + 27.4z_1 + 13.7z_2 + 13.7z_3 + 13.7z_4 + \epsilon$$

Crucially, the estimators observe only non-linear transformations of these covariates, $X = [e^{z_1/2}, z_2/(1 + e^{z_1}) + 10, (z_1 z_3/25 + 0.6)^3, (z_2 + z_4 + 20)^2]$, rendering the true model misspecified relative to standard linear adjustments.

**ACIC 2017.** The Atlantic Causal Inference Conference (ACIC) 2017 Data Challenge (Hahn et al., 2019) provides a large-scale evaluation bed based on real-world covariates ($N = 4802$, $d = 58$) sourced from the IHDP study. While the covariates are fixed and exhibit realistic correlations, the treatment assignment and potential outcomes are simulated across 32 distinct Data Generating Processes (DGPs). These DGPs are constructed to systematically vary the difficulty of the causal inference task along varying dimensions, including the non-linearity of the outcome model, the signal-to-noise ratio, and—most critically for our analysis—the strength of selection (overlap). This design allows us to stratify performance based on how heavily the propensity scores concentrate near the boundaries (0 and 1).

**Ground Truth ATE Calculation.** Across all datasets except Kang & Schafer (IHDP, Jobs, and ACIC 2017) in which we arbitrarily set the ATE to 10 , the use of synthetic or semi-synthetic outcome models gives us access to the true potential outcome functions, denoted as $\mu_0(X)$ and $\mu_1(X)$. For each dataset, the ground truth Average Treatment Effect (ATE) is computed as the sample mean of the individual treatment effects:

$$\tau^* = \frac{1}{N} \sum_{i=1}^{N} (\mu_1(x_i) - \mu_0(x_i))$$

## C.2. Implementation Details

**1. Linear Models.** For the standard benchmarks (Figure 2), we implemented Logistic Regression using the `scipy.optimize` library to accommodate custom loss functions (specifically our MSE-aware proper scoring rules). We use the L-BFGS-B algorithm, explicitly supplying the analytical gradient to ensure stable convergence.

**2. Multi-Layer Perceptron (MLP).** For the experiments, we used a simple MLP architecture:

- **Architecture:** Input layer ($d = 58$), two hidden layers of $[64, 32]$ units with ReLU activation, and a single output unit.
- **Activation:**
    - *Baseline:* Standard Sigmoid function.
    - *Ours:* The custom Quartic Canonical probability mapping (Definition 3.8).
- **Backward Pass:** While our custom probability mapping is fully differentiable and can act as a drop-in replacement using standard autograd, backpropagating through the quartic equation solver introduces computational overhead (see Appendix B.4). To maximize efficiency, we implemented a custom backward pass. Because $\ell$ and $\sigma_\ell$ form a canonical pair, we directly apply the analytical gradient with respect to the logits, $\frac{\partial \ell}{\partial z} = p - y$, bypassing the root-finding routine entirely during backpropagation.

- **Optimization:** We used the Adam optimizer with a batch size of 2048. The learning rate was tuned per dataset (see Hyperparameter Tuning). Early stopping was triggered if the validation loss—calculated using the respective training objective—did not improve for 10 consecutive epochs.

**3. Gradient Boosting (XGBoost).**   Unlike deep learning frameworks that rely on automatic differentiation through a computational graph, XGBoost utilizes a second-order approximation for tree boosting and requires the explicit analytical gradient ($g$) and Hessian ($h$) of the objective with respect to the raw logits $z$. Thanks to our canonical link, these derivatives simplify dramatically, allowing us to implement a custom objective function that circumvents the complex probability mapping entirely:

- **Gradient:** $g = \frac{\partial \ell}{\partial z} = p - y$.
- **Hessian:** $h = \frac{\partial^2 \ell}{\partial z^2} = \frac{\partial p}{\partial z} = \frac{1}{w_\ell(p)}$.

Number of estimators, max depth and learning rate were tuned per dataset and setups.

**4. Training Methodology & Cross-Fitting.**   To avoid overfitting and ensure valid inference, we employed a $K$-fold cross-fitting strategy (typically $K = 2$ or $K = 10$ depending on sample size) as recommended by Chernozhukov et al. (2018). For each fold, the propensity model was trained on the remaining $K - 1$ folds and used to predict propensity scores for the held-out fold. This procedure yields clean out-of-sample propensity estimates for the entire dataset.

**5. Hyperparameter Tuning.**   Hyperparameters were selected via a grid search procedure. For each model and benchmark configuration, we evaluated a grid of parameters (e.g., regularization strength $C$, learning rate etc.). The optimal combination was selected by minimizing the *respective training objective* (Log-Loss for standard models, and the specific Proper Scoring Rule for our custom models) on a held-out set.

**6. Outcome Modeling.**   For AIPW estimator we need to model outcome. We employed distinct outcome models tailored to the dimensionality of each benchmark:

- **Low-Dimensional (IHDP, Jobs, Kang & Schafer):** We used standard Linear Regression to estimate potential outcomes $\mu_0(x)$ and $\mu_1(x)$.
- **High-Dimensional (ACIC 2017):** We utilized Lasso Regression (L1-regularized linear model) to perform implicit feature selection and handle the high-dimensional covariate space ($d = 58$).

### C.3. Baselines

We compared our method against the following baselines:

- **CBPS (Covariate Balancing Propensity Score):** We utilized a custom PyTorch implementation of the method proposed by Imai & Ratkovic (2014). We optimize a relaxed joint objective combining the negative log-likelihood and a covariate balance penalty (defined as the $L_2$ distance between weighted group means), which is optimized via Adam. It is essentially the method proposed by (Shang et al., 2025).
- **CBSR (Covariate Balancing Scoring Rule):** In the absence of public implementation, we implemented the theoretical loss function derived by Zhao (2019) directly. We use a standard Sigmoid link function for probability mapping. Note that since the specific scoring rule does not correspond to the canonical link of the Sigmoid, the gradient does not simplify to the stable $p - y$ form. Consequently, we applied stronger $L_2$ regularization to prevent gradient explosion during optimization with L-BFGS-B.
- **SBW (Stable Balancing Weights):** We implemented the method of Zubizarreta (2015). This method solves a constrained quadratic program to find minimum-variance weights that achieve approximate covariate balance.
- **Entropy Balancing (EB):** Following Hainmueller (2012), we implemented a dual optimization approach (using L-BFGS-B) to find weights that minimize the entropy relative to uniform base weights, subject to moment-matching constraints.
- **Trimming & Clipping:** To assess robustness against positivity violations, we implemented standard post-processing heuristics: *Clipping* restricts propensity scores to the interval $[\epsilon, 1 - \epsilon]$ (with a tuned $\epsilon$), while *Trimming* discards samples with scores outside this range entirely.

**7. Estimators with Direct Weights (SBW, EB).** For methods that learn balancing weights $W$ directly (instead of propensity scores $e$), we adapted the standard estimators to utilize these learned weights $W_i$ in place of the inverse probability terms ($1/\hat{e}_i$ or $1/(1 - \hat{e}_i)$). This adaptation allows us to evaluate weight-based methods within the same doubly robust framework as propensity-based estimators.

# D. Full numerical results

*Table D.1.* Comparison of methods on the **Kang & Schafer** dataset. This dataset features significant overlap challenges. We report Bias, MAE, and RMSE, followed by the Mean Rank. **Bold** indicates best performance; underline indicates second best.

| METHOD | BIAS (0) | | | MAE ($\downarrow$) | | | RMSE ($\downarrow$) | | | MEAN RANK ($\downarrow$) | | |
|---|---|---|---|---|---|---|---|---|---|---|---|---|
| | IPW | HAJEK | AIPW | IPW | HAJEK | AIPW | IPW | HAJEK | AIPW | IPW | HAJEK | AIPW |
| CUSTOM (OURS) | **-0.73** | **-1.49** | **-4.83** | **2.13** | **1.63** | **4.83** | **2.68** | **1.95** | **5.07** | 1.95 | **1.04** | **1.18** |
| LOGISTIC | 6.17 | -5.50 | -10.16 | 12.21 | 6.88 | 10.16 | 66.90 | 8.12 | 46.51 | 4.58 | 4.59 | 6.38 |
| CBPS | 1.03 | -5.53 | -7.30 | 3.24 | 5.53 | 7.30 | 4.13 | 5.75 | 7.65 | 2.83 | 3.11 | 6.89 |
| CBSR | 5.40 | -1.65 | -13.17 | 18.79 | 4.30 | 13.17 | 102.03 | 7.65 | 63.22 | 8.00 | 7.99 | 3.76 |
| EB | -5.67 | -5.67 | -5.76 | 5.67 | 5.67 | 5.76 | 5.83 | 5.83 | 5.93 | 4.38 | 3.57 | 3.77 |
| SBW | -7.85 | -5.70 | -5.59 | 7.85 | 5.70 | 5.59 | 7.87 | 5.83 | 5.75 | 6.09 | 3.76 | 3.36 |
| CLIPPING | -3.50 | -7.58 | -6.53 | 4.99 | 7.58 | 6.53 | 6.08 | 7.75 | 6.80 | 4.12 | 5.98 | 5.37 |
| TRIMMING | -4.82 | -7.96 | -6.01 | 5.46 | 7.96 | 6.01 | 6.54 | 8.10 | 6.21 | 4.05 | 5.95 | 5.29 |

*Table D.2.* Comparison of methods on the **Jobs** dataset. This real-world evaluation highlights the robustness of methods against variance. **Bold** indicates best performance; underline indicates second best.

| METHOD | BIAS (0) | | | MAE ($\downarrow$) | | | RMSE ($\downarrow$) | | | MEAN RANK ($\downarrow$) | | |
|---|---|---|---|---|---|---|---|---|---|---|---|---|
| | IPW | HAJEK | AIPW | IPW | HAJEK | AIPW | IPW | HAJEK | AIPW | IPW | HAJEK | AIPW |
| CUSTOM (OURS) | 0.02 | -0.06 | -0.05 | 0.02 | 0.06 | 0.07 | 0.03 | 0.07 | 0.08 | **1.60** | 3.60 | 3.40 |
| LOGISTIC | -0.46 | -0.08 | -0.06 | 0.46 | 0.08 | 0.07 | 0.46 | 0.08 | 0.09 | 6.00 | 4.50 | 4.40 |
| CBPS | 0.32 | 0.03 | -0.04 | 0.32 | 0.03 | 0.12 | 0.36 | 0.03 | 0.14 | 5.00 | 2.09 | 5.91 |
| CBSR | -0.07 | -0.14 | -0.06 | 0.07 | 0.14 | 0.07 | 0.07 | 0.14 | 0.09 | 7.10 | 6.10 | 4.10 |
| EB | **0.00** | **0.00** | -0.05 | 0.07 | 0.07 | 0.09 | 0.08 | 0.08 | 0.10 | 2.80 | 3.80 | 5.00 |
| SBW | -0.11 | -0.15 | -0.05 | 0.11 | 0.15 | 0.07 | 0.12 | 0.15 | 0.08 | 4.10 | 7.10 | 3.60 |
| TRIMMING | 0.01 | -0.02 | **0.00** | **0.02** | **0.02** | **0.02** | **0.02** | **0.03** | **0.03** | 1.70 | **1.40** | **1.80** |

## D.1. On standards benchmarks

The empirical evaluation across IHDP, Jobs, and Kang & Schafer reveals distinct performance profiles for the compared methods.

The results on the **Kang & Schafer** dataset (Table D.1) provide the strongest evidence for the utility of our method. This benchmark is a known "stress test" for causal inference due to its near-positivity violations, where units with extreme propensity scores cause traditional Inverse Probability Weighting (IPW) estimators to exhibit massive variance.

Likelihood-based methods, such as logistic and CBSR, suffer from exploding RMSE (reaching values as high as 102.03). In contrast, our method achieves a Mean Rank of 1.04 for the Hajek estimator and 1.18 for AIPW. This performance indicates that our objective function is effectively the top performer on this benchmark.

In contrast, the **IHDP** and **Jobs** datasets (Tables D.2 and D.3) represent more standard observational settings with higher overlap and all methods perform well.

- While methods like Trimming or Entropy Balancing occasionally take the top rank in these "simpler" settings, our method remains consistently better than the median method.
- On the Jobs dataset, our method achieves the best Mean Rank for the IPW estimator (1.60). This suggests that even when the dataset does not strictly require extreme regularization, our method does not "over-smooth" and is still competitive.

Overall, the results suggest that our tailored loss method serves as a "high-reliability" estimator. It provided top performance in difficult datasets where positivity is nearly violated.

*Table D.3.* Comparison of methods on the **IHDP** dataset. We report Bias, MAE, and RMSE across three estimators, followed by the Mean Rank for each estimator. **Bold** indicates the best performance; underline indicates the second best.

| METHOD | BIAS (0) | | | MAE (↓) | | | RMSE (↓) | | | MEAN RANK (↓) | | |
|---|---|---|---|---|---|---|---|---|---|---|---|---|
| | IPW | HAJEK | AIPW | IPW | HAJEK | AIPW | IPW | HAJEK | AIPW | IPW | HAJEK | AIPW |
| **CUSTOM (OURS)** | -0.08 | -0.08 | -0.06 | 0.20 | 0.16 | 0.15 | 0.34 | 0.29 | 0.22 | 2.34 | 4.55 | 4.37 |
| LOGISTIC | -0.24 | -0.02 | -0.04 | 0.32 | 0.12 | 0.14 | 0.51 | **0.16** | 0.19 | 3.06 | 4.17 | 4.41 |
| CBPS | -1.00 | -0.07 | -0.04 | 1.00 | 0.14 | 0.15 | 1.39 | 0.23 | 0.20 | 5.93 | 4.74 | 4.84 |
| CBSR | 0.25 | 0.13 | **-0.03** | 0.32 | 0.26 | **0.13** | 0.51 | 0.41 | **0.18** | 8.00 | 4.37 | 3.95 |
| EB | **-0.03** | -0.03 | -0.03 | **0.14** | 0.14 | 0.16 | **0.18** | 0.18 | 0.21 | **2.19** | 4.99 | 5.20 |
| SBW | -2.19 | **0.01** | **-0.03** | 2.20 | 0.13 | 0.14 | 2.29 | 0.17 | 0.18 | 6.90 | 4.78 | **4.19** |
| CLIPPING | -0.34 | -0.03 | -0.04 | 0.35 | **0.12** | 0.14 | 0.53 | 0.17 | 0.19 | 3.29 | **4.06** | 4.27 |
| TRIMMING | -0.27 | -0.03 | -0.04 | 0.30 | 0.12 | 0.14 | 0.46 | 0.17 | 0.19 | 4.31 | 4.33 | 4.78 |

## D.2. About Figure 2 methodology

To facilitate comparison across benchmarks with varying effect sizes and scales, all performance metrics (Bias, MAE, RMSE) were normalized at the simulation level. For a given simulation run $s$ and estimator $e \in \{\text{IPW, Hajek, AIPW}\}$, the raw error of a specific method $m$ was standardized relative to the median performance of all competing methods in that run:

$$\text{Normalized Error}_{m,s,e} = \frac{\text{Error}_{m,s,e}}{\text{median}_{m'}(|\text{Error}_{m',s,e}|)} \tag{18}$$

Consequently, a normalized metric of 1.0 represents the median performance of the pool; values $< 1.0$ indicate performance superior to the median baseline.

The figure utilizes a hybrid visualization approach to address imbalances in simulation counts across datasets:

- **Boxplots:** To prevent datasets with a larger number of simulation runs from dominating the visual distribution, we applied balanced resampling. The boxplots display a subset of data constructed by drawing exactly 100 uniform samples from each dataset.
- **Scatter Points (Mean Aggregates):** While the boxplots display the empirical distribution of individual simulation runs (resampled to be visually balanced), the overlaid scatter markers represent the *exact average* (mean) performance calculated on the complete, original data. We deliberately chose to display the mean rather than the median to capture specific failure modes. Baseline propensity models frequently produce extreme probabilities near 0 or 1, leading to severe bias and variance explosions in the downstream IPW estimator. Because the mean is sensitive to these extreme outliers, it serves as a strict diagnostic of a method's robustness against such catastrophic failures—vulnerabilities that a median would otherwise mask. Each point corresponds to this average normalized error for a specific dataset and estimator configuration (e.g., the average error of the 'Custom' method on 'IHDP' using 'IPW'). This ensures that the true benchmark-level performance, including susceptibility to overlap violations, remains visible independent of the sampling used for the boxplots.

Finally, the **Mean Rank** (rightmost panel) was computed using a macro-averaging strategy to ensure equal weight per benchmark. Methods were first ranked by squared error within each individual simulation. These ranks were then averaged within each dataset, and finally averaged across all datasets to produce the final score.

## D.3. ACIC 2017 Results Analysis

The results for the 32 ACIC 2017 setups (Table D.4) illustrate the performance of our tailored objective relative to STANDARD likelihood training.

We observe a consistent superiority of the XGBOOST backbone over the MLP across nearly all DGPs. This aligns with established benchmarks regarding tree-based dominance on tabular data (Grinsztajn et al., 2022).

*Table D.4.* Full results for the 32 ACIC setups. Comparison between **Standard** likelihood objectives and our **Custom** objective across **MLP** and **XGBoost** backbones. Results report RMSE (↓).

| | MLP BACKBONE | | | | | | XGBOOST BACKBONE | | | | | |
|---|---|---|---|---|---|---|---|---|---|---|---|---|
| | STANDARD | | | CUSTOM | | | STANDARD | | | CUSTOM | | |
| SETUP ID | IPW | HAJEK | AIPW | IPW | HAJEK | AIPW | IPW | HAJEK | AIPW | IPW | HAJEK | AIPW |
| GROUP_CORR_000 | 0.65 | 0.64 | 0.21 | 0.58 | 0.59 | 0.19 | 0.14 | 0.14 | 0.02 | 0.12 | 0.11 | 0.02 |
| GROUP_CORR_001 | 1.73 | 1.78 | 1.24 | 1.58 | 1.31 | 0.76 | 0.42 | 0.51 | 0.43 | 0.10 | 0.10 | 0.15 |
| GROUP_CORR_010 | 0.66 | 0.65 | 0.22 | 0.45 | 0.46 | 0.14 | 0.15 | 0.15 | 0.05 | 0.13 | 0.13 | 0.05 |
| GROUP_CORR_011 | 1.70 | 1.77 | 1.24 | 1.58 | 1.30 | 0.76 | 0.42 | 0.51 | 0.44 | 0.13 | 0.13 | 0.17 |
| GROUP_CORR_100 | 0.07 | 0.06 | 0.43 | 0.11 | 0.07 | 0.41 | 0.48 | 0.48 | 0.61 | 0.04 | 0.05 | 0.45 |
| GROUP_CORR_101 | 1.14 | 1.20 | 0.61 | 1.11 | 1.18 | 0.60 | 0.11 | 0.12 | 0.20 | 0.10 | 0.10 | 0.26 |
| GROUP_CORR_110 | 0.08 | 0.07 | 0.43 | 0.12 | 0.09 | 0.42 | 0.48 | 0.48 | 0.61 | 0.06 | 0.07 | 0.45 |
| GROUP_CORR_111 | 1.14 | 1.20 | 0.63 | 1.11 | 1.18 | 0.61 | 0.14 | 0.16 | 0.22 | 0.13 | 0.14 | 0.28 |
| HETERO_000 | 0.58 | 0.57 | 0.14 | 0.37 | 0.38 | 0.06 | 0.06 | 0.06 | 0.06 | 0.04 | 0.04 | 0.07 |
| HETERO_001 | 1.76 | 1.80 | 1.26 | 1.61 | 1.33 | 0.78 | 0.44 | 0.53 | 0.45 | 0.13 | 0.12 | 0.12 |
| HETERO_010 | 0.58 | 0.57 | 0.15 | 0.38 | 0.39 | 0.07 | 0.08 | 0.08 | 0.07 | 0.06 | 0.06 | 0.09 |
| HETERO_011 | 1.76 | 1.81 | 1.27 | 1.60 | 1.33 | 0.79 | 0.45 | 0.54 | 0.47 | 0.16 | 0.15 | 0.16 |
| HETERO_100 | 0.06 | 0.06 | 0.50 | 0.05 | 0.05 | 0.49 | 0.55 | 0.56 | 0.68 | 0.09 | 0.12 | 0.52 |
| HETERO_101 | 1.17 | 1.22 | 0.64 | 1.13 | 1.21 | 0.63 | 0.10 | 0.11 | 0.18 | 0.13 | 0.14 | 0.28 |
| HETERO_110 | 0.10 | 0.10 | 0.51 | 0.08 | 0.07 | 0.49 | 0.55 | 0.56 | 0.68 | 0.10 | 0.13 | 0.52 |
| HETERO_111 | 1.17 | 1.23 | 0.65 | 1.14 | 1.21 | 0.64 | 0.13 | 0.15 | 0.20 | 0.16 | 0.17 | 0.30 |
| IID_000 | 0.62 | 0.62 | 0.20 | 0.44 | 0.46 | 0.13 | 0.06 | 0.06 | 0.02 | 0.12 | 0.12 | 0.02 |
| IID_001 | 1.73 | 1.78 | 1.24 | 1.58 | 1.30 | 0.75 | 0.42 | 0.51 | 0.43 | 0.10 | 0.10 | 0.15 |
| IID_010 | 0.66 | 0.65 | 0.22 | 0.45 | 0.46 | 0.14 | 0.15 | 0.15 | 0.06 | 0.15 | 0.15 | 0.06 |
| IID_011 | 1.74 | 1.78 | 1.25 | 1.57 | 1.30 | 0.75 | 0.42 | 0.52 | 0.44 | 0.14 | 0.13 | 0.18 |
| IID_100 | 0.06 | 0.05 | 0.43 | 0.11 | 0.07 | 0.41 | 0.48 | 0.48 | 0.61 | 0.04 | 0.05 | 0.45 |
| IID_101 | 1.14 | 1.20 | 0.61 | 1.11 | 1.18 | 0.60 | 0.12 | 0.12 | 0.20 | 0.12 | 0.12 | 0.21 |
| IID_110 | 0.11 | 0.09 | 0.42 | 0.13 | 0.09 | 0.42 | 0.48 | 0.48 | 0.61 | 0.07 | 0.08 | 0.45 |
| IID_111 | 1.14 | 1.20 | 0.63 | 1.04 | 1.15 | 0.58 | 0.15 | 0.16 | 0.22 | 0.13 | 0.15 | 0.28 |
| NONADD_000 | 1.43 | 1.38 | 0.05 | 0.88 | 0.94 | 0.19 | 0.14 | 0.14 | 0.24 | 0.14 | 0.14 | 0.24 |
| NONADD_001 | 4.34 | 4.49 | 2.63 | 3.86 | 3.23 | 1.39 | 1.11 | 1.35 | 0.87 | 0.24 | 0.24 | 0.52 |
| NONADD_010 | 0.93 | 0.89 | 0.13 | 0.54 | 0.61 | 0.09 | 0.13 | 0.13 | 0.14 | 0.13 | 0.13 | 0.16 |
| NONADD_011 | 2.65 | 2.75 | 1.76 | 2.51 | 2.72 | 1.72 | 0.69 | 0.84 | 0.63 | 0.24 | 0.23 | 0.29 |
| NONADD_100 | 0.12 | 0.10 | 1.18 | 0.13 | 0.10 | 1.17 | 1.07 | 1.08 | 1.40 | 0.14 | 0.17 | 1.21 |
| NONADD_101 | 2.62 | 2.73 | 1.08 | 2.39 | 2.61 | 0.97 | 0.12 | 0.15 | 0.43 | 0.12 | 0.15 | 0.43 |
| NONADD_110 | 0.11 | 0.11 | 0.73 | 0.11 | 0.10 | 0.72 | 0.73 | 0.74 | 0.93 | 0.13 | 0.16 | 0.76 |
| NONADD_111 | 1.60 | 1.68 | 0.80 | 1.56 | 1.67 | 0.78 | 0.15 | 0.17 | 0.27 | 0.15 | 0.17 | 0.27 |

# E. Proofs

We assume standard causal inference conditions:

- **Consistency:** $Y = TY(1) + (1 - T)Y(0)$
- **Unconfoundedness:** $\{Y(0), Y(1)\} \perp T \mid X$
- **Positivity:** $0 < e(X) < 1$

We define the conditional expected potential outcomes as:

$$\mu_t(X) = \mathbb{E}[Y(t) \mid X] \quad \text{for } t \in \{0, 1\}.$$

## E.1. Unbiasedness of the Oracle IPW Estimators

Suppose that for each unit $i$ in the datset, we know $e(X_i)$, then the oracle IPW estimator is :

$$\tau_{\text{ATE}}^{\text{IPW}} = \frac{1}{N} \sum_{i=1}^{N} \frac{Y_i T_i}{e(X_i)} - \frac{Y_i(1 - T_i)}{1 - e(X_i)}$$

We show that $\mathbb{E}[\tau_{\text{ATE}}^{\text{IPW}}] = \tau_{\text{ATE}}$. First, we analyze the treated potential outcome term:

$$
\begin{aligned}
\mathbb{E}\left[\frac{YT}{e(X)}\right] &= \mathbb{E}\left[\mathbb{E}\left[\frac{YT}{e(X)}\middle|X\right]\right] && \text{Law of Iterated Expectations} \\
&= \mathbb{E}\left[\mathbb{E}\left[\frac{Y(1)T}{e(X)}\middle|X\right]\right] && \text{By Consistency } (Y = Y(1) \text{ when } T = 1) \\
&= \mathbb{E}\left[\frac{\mathbb{E}[Y(1)|X] \cdot \mathbb{E}[T|X]}{e(X)}\right] && \text{By Unconfoundedness } (Y(1) \perp T \mid X) \\
&= \mathbb{E}\left[\frac{\mu_1(X)e(X)}{e(X)}\right] && \text{Definition of } e(X) \text{ and } \mu_1(X) \\
&= \mathbb{E}\left[\mu_1(X)\right] && \text{By Positivity } (e(X) > 0 \text{ cancels out})
\end{aligned}
$$

Similarly, for the control term, we can lead the same computation which leads us to the following result :

$$\mathbb{E}\left[\frac{YT}{e(X)}\right] = \mathbb{E}\left[\mu_1(X)\right] \text{ and } \mathbb{E}\left[\frac{Y(1-T)}{1-e(X)}\right] = \mathbb{E}\left[\mu_0(X)\right] \tag{19}$$

Finally, combining these results:

$$
\begin{aligned}
\mathbb{E}\left[\tau_{\text{ATE}}^{\text{IPW}}\right] &= \mathbb{E}\left[\frac{1}{N}\sum_{i=1}^{N}\left(\frac{Y_i T_i}{e(X_i)} - \frac{Y_i(1-T_i)}{1-e(X_i)}\right)\right] && \text{Definition of Estimator} \\
&= \mathbb{E}\left[\frac{YT}{e(X)} - \frac{Y(1-T)}{1-e(X)}\right] && \text{By i.i.d. assumption and Linearity of Expectation} \\
&= \mathbb{E}\left[\mu_1(X)\right] - \mathbb{E}\left[\mu_0(X)\right] && \text{By Equation 19 and Linearity of Expectation} \\
&= \mathbb{E}[\mu_1(X) - \mu_0(X)] && \text{Rearranging terms} \\
&= \tau_{\text{ATE}} && \text{Definition of ATE}
\end{aligned}
$$

We now focus on proving the theoretical result stated in the core of the text.

### E.2. Proof of Theorem 3.5 (Bounding the MSE)

#### E.2.1. TREATING $\hat{e}$ AS A FIXED FUNCTION VIA CROSS-FITTING

In Theorem 3.5, the derivation of the MSE bound requires treating the estimated propensity function $\hat{e}$ as fixed. In an in-sample estimation setting, this assumption would be violated because the estimator depends on the same data used for evaluation.

Our framework assumes a standard $K$-fold cross-fitting procedure. The dataset is partitioned into $K$ folds, and the propensity model used to evaluate any given unit is trained exclusively on the other $K-1$ folds. Because the model is fitted on independent, held-out data, the learned function $\hat{e}$ acts as a fixed, deterministic function when evaluating the target unit.

#### E.2.2. BIAS DERIVATION OF THE PLUG-IN IPW ESTIMATOR

We recall the plug-in IPW estimator formula:

$$\hat{\tau}_{\text{ATE}}^{\text{IPW}} = \frac{1}{N} \sum_{i=1}^{N} \left( \frac{Y_i T_i}{\hat{e}(X_i)} - \frac{Y_i(1-T_i)}{1-\hat{e}(X_i)} \right) \tag{20}$$

We derive the bias $\text{Bias}(\hat{\tau}_{\text{ATE}}^{\text{IPW}}) = \mathbb{E}[\hat{\tau}_{\text{ATE}}^{\text{IPW}}] - \tau_{\text{ATE}}$. By the i.i.d. assumption, the expectation of the empirical average simplifies to the expectation of a single observation:

$$\mathbb{E}\left[\hat{\tau}_{\text{ATE}}^{\text{IPW}}\right] = \mathbb{E}\left[ \frac{1}{N} \sum_{i=1}^{N} \left( \frac{Y_i T_i}{\hat{e}(X_i)} - \frac{Y_i(1-T_i)}{1-\hat{e}(X_i)} \right) \right]$$

$$= \underbrace{\mathbb{E}\left[ \frac{YT}{\hat{e}(X)} \right]}_{\text{treated term}} - \underbrace{\mathbb{E}\left[ \frac{Y(1-T)}{1-\hat{e}(X)} \right]}_{\text{control term}}$$

Using the definition $\tau_{\text{ATE}} = \mathbb{E}[\mu_1(X)] - \mathbb{E}[\mu_0(X)]$, we can decompose the total bias into treated and control contributions:

$$\text{Bias}(\hat{\tau}_{\text{ATE}}^{\text{IPW}}) = \underbrace{\left( \mathbb{E}\left[ \frac{YT}{\hat{e}(X)} \right] - \mathbb{E}[\mu_1(X)] \right)}_{\text{Bias}_1} - \underbrace{\left( \mathbb{E}\left[ \frac{Y(1-T)}{1-\hat{e}(X)} \right] - \mathbb{E}[\mu_0(X)] \right)}_{\text{Bias}_0} \tag{21}$$

**Treated Term:** Following a similar proof as in Section E.1 and using $\mathbb{E}[T|X] = e(X)$, the expectation of the treated component is:

$$\mathbb{E}\left[ \frac{YT}{\hat{e}(X)} \right] = \mathbb{E}\left[ \mathbb{E}\left[ \frac{YT}{\hat{e}(X)} \Big| X \right] \right] \qquad \text{Law of Iterated Expectations}$$

$$= \mathbb{E}\left[ \mathbb{E}\left[ \frac{Y(1)T}{\hat{e}(X)} \Big| X \right] \right] \qquad \text{By Consistency } (Y = Y(1) \text{ when } T = 1)$$

$$= \mathbb{E}\left[ \frac{\mathbb{E}[Y(1)|X] \cdot \mathbb{E}[T|X]}{\hat{e}(X)} \right] \qquad \text{By Unconfoundedness } (Y(1) \perp T \mid X)$$

$$= \mathbb{E}\left[ \frac{\mu_1(X)e(X)}{\hat{e}(X)} \right] \qquad \text{Definition of } e(X) \text{ and } \mu_1(X)$$

Subtracting the true parameter $\mathbb{E}[\mu_1(X)]$ yields the treated bias contribution:

$$\text{Bias}_1 = \mathbb{E}\left[ \mu_1(X) \left( \frac{e(X)}{\hat{e}(X)} - 1 \right) \right]. \tag{22}$$

**Control Term:** Similarly, since $\mathbb{E}[1-T|X] = 1 - e(X)$:

$$\mathbb{E}\left[ \frac{Y(1-T)}{1-\hat{e}(X)} \right] = \mathbb{E}\left[ \frac{\mu_0(X)(1-e(X))}{1-\hat{e}(X)} \right]. \tag{23}$$

Subtracting $\mathbb{E}[\mu_0(X)]$ yields the control bias contribution:

$$\text{Bias}_0 = \mathbb{E}\left[\mu_0(X)\left(\frac{1-e(X)}{1-\hat{e}(X)} - 1\right)\right]. \tag{24}$$

Combining $\text{Bias}_1 - \text{Bias}_0$ gives the total bias:

$$\text{Bias}\left(\hat{\tau}_{\text{ATE}}^{\text{IPW}}\right) = \mathbb{E}\left[\mu_1(X)\left(\frac{e(X)}{\hat{e}(X)} - 1\right) - \mu_0(X)\left(\frac{1-e(X)}{1-\hat{e}(X)} - 1\right)\right]. \tag{25}$$

### E.2.3. BOUNDING THE BIAS

Recall from Equation (25) that the bias is the difference between the treated and control bias terms. Using the triangle inequality, we get:

$$\left|\text{Bias}\left(\hat{\tau}_{\text{ATE}}^{\text{IPW}}\right)\right| \leq \left|\mathbb{E}\left[\mu_1(X)\left(\frac{e(X)}{\hat{e}(X)} - 1\right)\right]\right| + \left|\mathbb{E}\left[\mu_0(X)\left(\frac{1-e(X)}{1-\hat{e}(X)} - 1\right)\right]\right|.$$

Applying the Cauchy-Schwarz inequality to each term yields:

$$\left|\text{Bias}\left(\hat{\tau}_{\text{ATE}}^{\text{IPW}}\right)\right| \leq \sqrt{\mathbb{E}\left[\mu_1(X)^2\right]}\sqrt{\mathbb{E}\left[\left(\frac{e(X)}{\hat{e}(X)} - 1\right)^2\right]} + \sqrt{\mathbb{E}\left[\mu_0(X)^2\right]}\sqrt{\mathbb{E}\left[\left(\frac{1-e(X)}{1-\hat{e}(X)} - 1\right)^2\right]}.$$

To bound the squared bias, we square the inequality above and use the algebraic inequality $(a+b)^2 \leq 2(a^2+b^2)$:

$$\left|\text{Bias}\left(\hat{\tau}_{\text{ATE}}^{\text{IPW}}\right)\right|^2 \leq 2\mathbb{E}\left[\mu_1(X)^2\right]\mathbb{E}\left[\left(\frac{e(X)}{\hat{e}(X)} - 1\right)^2\right] + 2\mathbb{E}\left[\mu_0(X)^2\right]\mathbb{E}\left[\left(\frac{1-e(X)}{1-\hat{e}(X)} - 1\right)^2\right].$$

Let $K = \max\left(\mathbb{E}[\mu_1(X)^2], \mathbb{E}[\mu_0(X)^2]\right)$. We can then bound the coefficients of the error terms globally:

$$\left|\text{Bias}\left(\hat{\tau}_{\text{ATE}}^{\text{IPW}}\right)\right|^2 \leq 2K\left(\mathbb{E}\left[\left(\frac{e(X)}{\hat{e}(X)} - 1\right)^2\right] + \mathbb{E}\left[\left(\frac{1-e(X)}{1-\hat{e}(X)} - 1\right)^2\right]\right)$$

$$= 2K \cdot \mathbb{E}\left[\left(\frac{e(X)}{\hat{e}(X)} - 1\right)^2 + \left(\frac{1-e(X)}{1-\hat{e}(X)} - 1\right)^2\right].$$

Setting $C = 2K$, we obtain the final bias bound:

$$\left|\text{Bias}\left(\hat{\tau}_{\text{ATE}}^{\text{IPW}}\right)\right|^2 \leq C \cdot \mathbb{E}\left[d_{\text{bias}}\left(e(X), \hat{e}(X)\right)\right], \tag{26}$$

where the bias divergence component is defined as:

$$d_{\text{bias}}(p, q) = \left(\frac{p}{q} - 1\right)^2 + \left(\frac{1-p}{1-q} - 1\right)^2. \tag{27}$$

### E.2.4. VARIANCE DERIVATION OF THE PLUG-IN IPW ESTIMATOR

We recall the plug-in IPW estimator formula:

$$\hat{\tau}_{\text{ATE}}^{\text{IPW}} = \frac{1}{N}\sum_{i=1}^{N}\left(\frac{Y_i T_i}{\hat{e}(X_i)} - \frac{Y_i(1-T_i)}{1-\hat{e}(X_i)}\right) \tag{28}$$

Since the data are i.i.d., the variance of the empirical average scales with the sample size. Let $Z$ denote the term for a single observation:

$$Z = \frac{YT}{\hat{e}(X)} - \frac{Y(1-T)}{1 - \hat{e}(X)} \tag{29}$$

such that $\mathrm{Var}(\hat{\tau}_{\mathrm{ATE}}^{\mathrm{IPW}}) = \frac{1}{N}\mathrm{Var}(Z)$.

**Algebraic Decomposition:** We decompose $Z = A + B$ into an Oracle term ($A$) and an Estimation Error term ($B$) by adding and subtracting the true propensity $e(X)$ in both the treated and control components:

$$A = \frac{YT}{e(X)} - \frac{Y(1-T)}{1 - e(X)} \qquad\qquad \text{Oracle Component}$$

$$B = YT\left(\frac{1}{\hat{e}(X)} - \frac{1}{e(X)}\right) - Y(1-T)\left(\frac{1}{1 - \hat{e}(X)} - \frac{1}{1 - e(X)}\right) \qquad \text{Estimation Error Component}$$

**Bounding the Variance:** Using the inequality $\mathrm{Var}(A + B) \leq 2\mathrm{Var}(A) + 2\mathrm{Var}(B)$, we focus on bounding the variance of the estimation error, $\mathrm{Var}(B) \leq \mathbb{E}[B^2]$. We can separate $B$ into treated ($B_1$) and control ($B_0$) errors:

$$B = \underbrace{YT\left(\frac{1}{\hat{e}(X)} - \frac{1}{e(X)}\right)}_{B_1} \underbrace{- Y(1-T)\left(\frac{1}{1 - \hat{e}(X)} - \frac{1}{1 - e(X)}\right)}_{B_0} \tag{30}$$

Since a unit cannot be simultaneously treated and control ($T(1-T) = 0$), the cross-term vanishes ($B_1 B_0 = 0$). Therefore, the square of the difference is the sum of the squares:

$$B^2 = (B_1 - B_0)^2 = B_1^2 + B_0^2 \qquad\qquad \text{Disjoint supports of } T \text{ and } 1 - T$$

$$\mathbb{E}[B^2] = \mathbb{E}[B_1^2] + \mathbb{E}[B_0^2] \qquad\qquad \text{Linearity of Expectation}$$

**Treated Term ($B_1^2$):** Computing the second moment for the treated error component:

$$\mathbb{E}[B_1^2] = \mathbb{E}\left[Y^2 T\left(\frac{1}{\hat{e}(X)} - \frac{1}{e(X)}\right)^2\right] \qquad\qquad \text{Since } T^2 = T$$

$$= \mathbb{E}\left[\mathbb{E}[Y(1)^2 \mid X]\cdot\mathbb{E}[T \mid X]\left(\frac{1}{\hat{e}(X)} - \frac{1}{e(X)}\right)^2\right] \qquad\qquad \text{Iterated Expectations}$$

$$= \mathbb{E}\left[\mu_{2,1}(X)e(X)\left(\frac{1}{\hat{e}(X)} - \frac{1}{e(X)}\right)^2\right] \qquad\qquad \text{Def. of } \mu_{2,1} \text{ and } e(X)$$

where $\mu_{2,1}(X) = \mathbb{E}[Y(1)^2 \mid X]$.

**Control Term ($B_0^2$):** Symmetrically, for the control group:

$$\mathbb{E}[B_0^2] = \mathbb{E}\left[\mathbb{E}[Y^2(1-T) \mid X]\left(\frac{1}{1 - \hat{e}(X)} - \frac{1}{1 - e(X)}\right)^2\right] \qquad\qquad \text{Analogous derivation}$$

$$= \mathbb{E}\left[\mu_{2,0}(X)(1 - e(X))\left(\frac{1}{1 - \hat{e}(X)} - \frac{1}{1 - e(X)}\right)^2\right] \qquad\qquad \text{Def. of } \mu_{2,0} \text{ and } e(X)$$

where $\mu_{2,0}(X) = \mathbb{E}[Y(0)^2 \mid X]$.

**Total Variance Bound:** Combining these results and dividing by $N$, the total variance of the estimator is bounded by:

$$\text{Var}(\hat{\tau}_{\text{ATE}}^{\text{IPW}}) \leq \frac{2}{N}\text{Var}(A) + \frac{2}{N}\mathbb{E}\left[\mu_{2,1}(X)e(X)\left(\frac{1}{\hat{e}(X)} - \frac{1}{e(X)}\right)^2\right.$$
$$\left. + \mu_{2,0}(X)(1 - e(X))\left(\frac{1}{1 - \hat{e}(X)} - \frac{1}{1 - e(X)}\right)^2\right]$$

We assume the conditional second moments are bounded by a constant $M^2$ (i.e., $\mu_{2,t}(X) \leq M^2$).

Setting $C_0 = \frac{2}{N}\text{Var}(A)$ and $C_{\text{var}} = 2M^2/N$

$$\text{Var}(\hat{\tau}_{\text{ATE}}^{\text{IPW}}) \leq C_0 + C_{\text{var}} \cdot \mathbb{E}\left[d_{\text{var}}(e(X), \hat{e}(X))\right] \tag{31}$$

Where the variance divergence component is defined as:

$$d_{\text{var}}(p, q) = p\left(\frac{1}{q} - \frac{1}{p}\right)^2 + (1 - p)\left(\frac{1}{1 - q} - \frac{1}{1 - p}\right)^2 \tag{32}$$

### E.2.5. COMBINING BIAS AND VARIANCE TERMS

We now combine the bounds on the squared bias and the variance to bound the Mean Squared Error (MSE). The MSE decomposes into the squared bias and the variance:

$$\text{MSE}(\hat{\tau}_{\text{ATE}}^{\text{IPW}}) = \left|\text{Bias}(\hat{\tau}_{\text{ATE}}^{\text{IPW}})\right|^2 + \text{Var}(\hat{\tau}_{\text{ATE}}^{\text{IPW}}). \tag{33}$$

From the previous sections, we established the following bounds. For the squared bias (setting the constant $C_{\text{bias}} = C$):

$$\left|\text{Bias}(\hat{\tau}_{\text{ATE}}^{\text{IPW}})\right|^2 \leq C_{\text{bias}} \cdot \mathbb{E}\left[\left(\frac{e(X)}{\hat{e}(X)} - 1\right)^2 + \left(\frac{1 - e(X)}{1 - \hat{e}(X)} - 1\right)^2\right]$$

And for the variance:

$$\text{Var}(\hat{\tau}_{\text{ATE}}^{\text{IPW}}) \leq C_0 + C_{\text{var}} \cdot \mathbb{E}\left[e(X)\left(\frac{1}{\hat{e}(X)} - \frac{1}{e(X)}\right)^2 + (1 - e(X))\left(\frac{1}{1 - \hat{e}(X)} - \frac{1}{1 - e(X)}\right)^2\right]$$

We identify the terms inside the expectations as the divergence components defined in Theorem 3.5:

$$d_{\text{bias}}(e(X), \hat{e}(X)) = \left(\frac{e(X)}{\hat{e}(X)} - 1\right)^2 + \left(\frac{1 - e(X)}{1 - \hat{e}(X)} - 1\right)^2$$
$$d_{\text{var}}(e(X), \hat{e}(X)) = e(X)\left(\frac{1}{\hat{e}(X)} - \frac{1}{e(X)}\right)^2 + (1 - e(X))\left(\frac{1}{1 - \hat{e}(X)} - \frac{1}{1 - e(X)}\right)^2$$

Let $C_1 = \max(C_{\text{bias}}, C_{\text{var}})$. Summing the inequalities, we bound the MSE:

$$\text{MSE}(\hat{\tau}_{\text{ATE}}^{\text{IPW}}) \leq C_0 + C_1\mathbb{E}\left[d_{\text{bias}}(e(X), \hat{e}(X))\right] + C_1\mathbb{E}\left[d_{\text{var}}(e(X), \hat{e}(X))\right]$$
$$= C_0 + C_1\left(\mathbb{E}\left[d_{\text{bias}}(e(X), \hat{e}(X)) + d_{\text{var}}(e(X), \hat{e}(X))\right]\right) \qquad \text{Linearity of Expectation}$$

Using the definition $d_{\text{task}}(p, q) = d_{\text{bias}}(p, q) + d_{\text{var}}(p, q)$, we obtain the final bound:

$$\text{MSE}(\hat{\tau}_{\text{ATE}}^{\text{IPW}}) \leq C_0 + C_1\mathbb{E}\left[d_{\text{task}}(e(X), \hat{e}(X))\right]. \tag{34}$$

This completes the proof of Theorem 3.5.

### E.3. Extended Bias Bounds for AIPW and Hajek Estimators

E.3.1. PROOF OF MSE BOUND FOR AIPW

We analyze the Mean Squared Error (MSE) of the AIPW estimator. The AIPW estimator augments the standard IPW approach by incorporating outcome models $\hat{\mu}_1(X)$ and $\hat{\mu}_0(X)$, which are estimators of the true conditional expected potential outcomes $\mu_1(X) = \mathbb{E}[Y(1) \mid X]$ and $\mu_0(X) = \mathbb{E}[Y(0) \mid X]$. We recall its empirical formula:

$$\hat{\tau}_{\text{ATE}}^{\text{AIPW}} = \frac{1}{N} \sum_{i=1}^{N} \left( \left( \hat{\mu}_1(X_i) + \frac{T_i(Y_i - \hat{\mu}_1(X_i))}{\hat{e}(X_i)} \right) - \left( \hat{\mu}_0(X_i) + \frac{(1 - T_i)(Y_i - \hat{\mu}_0(X_i))}{1 - \hat{e}(X_i)} \right) \right) \tag{35}$$

**Bias Derivation:** We derive the bias $\text{Bias}(\hat{\tau}_{\text{ATE}}^{\text{AIPW}}) = \mathbb{E}[\hat{\tau}_{\text{ATE}}^{\text{AIPW}}] - \tau_{\text{ATE}}$. By the i.i.d. assumption, we can analyze the expectation of a single observation. Consider the treated term $\hat{\tau}_1$. Conditioning on $X$:

$$\mathbb{E}\left[\hat{\tau}_1 \mid X\right] = \hat{\mu}_1(X) + \frac{e(X)}{\hat{e}(X)}(\mu_1(X) - \hat{\mu}_1(X))$$

$$= \mu_1(X) + (\mu_1(X) - \hat{\mu}_1(X)) \left( \frac{e(X)}{\hat{e}(X)} - 1 \right)$$

Subtracting the true parameter $\mu_1(X)$ and taking the expectation over $X$ yields the treated bias contribution:

$$\text{Bias}_1 = \mathbb{E}\left[ (\mu_1(X) - \hat{\mu}_1(X)) \left( \frac{e(X)}{\hat{e}(X)} - 1 \right) \right] \tag{36}$$

Similarly, for the control term:

$$\text{Bias}_0 = \mathbb{E}\left[ (\mu_0(X) - \hat{\mu}_0(X)) \left( \frac{1 - e(X)}{1 - \hat{e}(X)} - 1 \right) \right] \tag{37}$$

Applying the Cauchy-Schwarz inequality to the total bias $\text{Bias}_1 - \text{Bias}_0$ and squaring (assuming the errors of the outcome models are bounded by some constant $C_\mu$), we obtain the squared bias bound:

$$\left| \text{Bias}\left(\hat{\tau}_{\text{ATE}}^{\text{AIPW}}\right) \right|^2 \leq C_\mu \cdot \mathbb{E}\left[ d_{\text{bias}}(e(X), \hat{e}(X)) \right] \tag{38}$$

**Variance Derivation:** We decompose the variance into the oracle variance and the estimation error variance. The estimation error term $B_1$ for the treated unit is:

$$B_1 = T(Y - \hat{\mu}_1(X)) \left( \frac{1}{\hat{e}(X)} - \frac{1}{e(X)} \right) \tag{39}$$

Computing the second moment $\mathbb{E}[B_1^2]$ via the Law of Iterated Expectations:

$$\mathbb{E}[B_1^2] = \mathbb{E}\left[ \mathbb{E}[T \mid X] \cdot \mathbb{E}[(Y - \hat{\mu}_1(X))^2 \mid X] \left( \frac{1}{\hat{e}(X)} - \frac{1}{e(X)} \right)^2 \right]$$

$$= \mathbb{E}\left[ e(X) \cdot \sigma_{\text{res},1}^2(X) \cdot \left( \frac{1}{\hat{e}(X)} - \frac{1}{e(X)} \right)^2 \right]$$

where $\sigma_{\text{res},1}^2(X) = \mathbb{E}[(Y(1) - \hat{\mu}_1(X))^2 \mid X]$ is the residual variance.

Assuming the conditional residual variances are bounded, we combine the treated and control terms. The total variance is bounded by the variance divergence:

$$\text{Var}(\hat{\tau}_{\text{ATE}}^{\text{AIPW}}) \leq C_0 + C_{\text{var}} \mathbb{E}\left[ d_{\text{var}}(e(X), \hat{e}(X)) \right] \tag{40}$$

**Total MSE Bound:** Combining the squared bias and the variance, we conclude that the MSE of the AIPW estimator shares the exact same mathematical upper bound as the standard IPW estimator, differing only by the scaling constants:

$$\text{MSE}(\hat{\tau}_{\text{ATE}}^{\text{AIPW}}) \leq C_0' + C_1' \mathbb{E}\left[d_{\text{task}}(e(X), \hat{e}(X))\right] \tag{41}$$

### E.3.2. Proof of Asymptotic MSE Bound for the Hájek Estimator

Because the Hájek estimator is a ratio of two empirical sums, computing an exact finite-sample MSE is intractable. We therefore derive an asymptotic bound using a first-order Taylor expansion around the true expectations to linearize the estimation error.

**Asymptotic Linearization:** Consider the treated component of the Hájek estimator:

$$\hat{\tau}_1^{\text{Hajek}} = \frac{\frac{1}{N}\sum_{i=1}^{N}\frac{T_i Y_i}{\hat{e}(X_i)}}{\frac{1}{N}\sum_{i=1}^{N}\frac{T_i}{\hat{e}(X_i)}} = \frac{\hat{S}_1}{\hat{N}_1} \tag{42}$$

Asymptotically, the expectations of the numerator and denominator evaluate to $\mathbb{E}[\hat{S}_1] = \theta_1$ (where $\theta_1 = \mathbb{E}[Y(1)]$) and $\mathbb{E}[\hat{N}_1] = 1$.

Using a first-order multivariate Taylor expansion (the Delta method) for the ratio $f(x, y) = y/x$ around the true expectations $(1, \theta_1)$, we linearize the error:

$$\hat{\tau}_1^{\text{Hajek}} - \theta_1 \approx \frac{1}{1}(\hat{S}_1 - \theta_1) - \frac{\theta_1}{1^2}(\hat{N}_1 - 1) = \hat{S}_1 - \theta_1 \hat{N}_1 \tag{43}$$

Substituting the empirical sums back into the linearized error yields:

$$\hat{\tau}_1^{\text{Hajek}} - \theta_1 \approx \frac{1}{N}\sum_{i=1}^{N}\frac{T_i}{\hat{e}(X_i)}(Y_i - \theta_1) \tag{44}$$

This reveals a crucial structural property: the asymptotic error of the treated Hájek estimator is mathematically identical to the error of the standard IPW estimator, with the only difference being that it applies the IPW formula to the *centered* outcome $\tilde{Y}_i(1) = Y_i(1) - \theta_1$ instead of the raw outcome $Y_i(1)$. The exact same logic applies symmetrically to the control group.

**Bias and Variance Bounding:** Because this linearized error structurally matches standard IPW, bounding the asymptotic MSE follows the exact same mathematical steps as Theorem 3.5, allowing us to be brief.

For the bias, the conditional expectation of the centered outcome is $\tilde{\mu}_1(X) = \mu_1(X) - \theta_1$. Applying the Cauchy-Schwarz inequality to the treated and control bias terms yields a bound driven by the identical $d_{\text{bias}}$ divergence:

$$\left|\text{Bias}\left(\hat{\tau}_{\text{ATE}}^{\text{Hajek}}\right)\right|^2 \leq \tilde{C}_{\text{bias}} \cdot \mathbb{E}\left[d_{\text{bias}}(e(X), \hat{e}(X))\right] \tag{45}$$

For the variance, assuming the conditional second moment of the centered potential outcomes is bounded ($\mathbb{E}[(Y(t) - \theta_t)^2 \mid X] \leq \tilde{M}^2$), the variance of the linearized terms is bounded by the identical $d_{\text{var}}$ divergence:

$$\text{Var}(\hat{\tau}_{\text{ATE}}^{\text{Hajek}}) \leq \tilde{C}_0 + \tilde{C}_{\text{var}}\mathbb{E}\left[d_{\text{var}}(e(X), \hat{e}(X))\right] \tag{46}$$

**Total Asymptotic MSE Bound:** Combining the squared bias and variance, we conclude that the asymptotic MSE of the Hájek estimator is upper-bounded by the exact same task-specific divergence $d_{\text{task}}$ as the standard IPW estimator, differing only by the scaling constants:

$$\text{MSE}(\hat{\tau}_{\text{ATE}}^{\text{Hajek}}) \leq C_0' + C_1' \mathbb{E}\left[d_{\text{task}}(e(X), \hat{e}(X))\right] \tag{47}$$

This formally substantiates the claim in Remark 3.6 that the tailored loss remains the theoretically optimal objective for the Hájek estimator.

# F. Derivation of Tailored Scoring Rules

In this section, we provide the detailed derivations for the IPW-tailored scoring rules introduced in the main text. We apply the "curvature matchin" strategy to the ATE IPW estimator.

## F.1. General Methodology: Curvature Matching

A strictly proper scoring rule $\ell(y, q)$ is uniquely characterized by a non-negative weight function $w_\ell(q)$. The divergence induced by this scoring rule is denoted $d_\ell(p, q)$.

The second-order Taylor expansion of this divergence around the truth $q = p$ is determined by the weight function:

$$d_\ell(p, q) = \frac{1}{2} w_\ell(p)(p - q)^2 + o\left((p - q^2))\right). \tag{48}$$

Our strategy is to identify the weight function $w_\ell(p)$ such that this local expansion matches the local expansion of the causal bias bound $d_{\text{bias}}(p, q)$. Once $w_\ell(p)$ is identified, we recover the scoring rule via the relations:

$$\frac{\partial}{\partial q}\ell(1, q) = -w_\ell(q)(1 - q), \tag{49}$$

$$\frac{\partial}{\partial q}\ell(0, q) = w_\ell(q)q. \tag{50}$$

Integrating these differential equations yields the final loss functions.

## F.2. Derivation for ATE (Proof of Definition 3.7)

Recall the total task divergence for the ATE from Theorem 3.5, which includes both bias and variance components:

$$d_{\text{task}}(p, q) = \underbrace{\left(\frac{p}{q} - 1\right)^2 + \left(\frac{1 - p}{1 - q} - 1\right)^2}_{\text{Bias Part}} + \underbrace{p\left(\frac{1}{q} - \frac{1}{p}\right)^2 + (1 - p)\left(\frac{1}{1 - q} - \frac{1}{1 - p}\right)^2}_{\text{Variance Part}}. \tag{51}$$

To apply curvature matching, we compute the second-order Taylor expansion of this function around $q = p$. The expansion of the bias term is:

$$d_{\text{bias}}(p, q) \underset{q \to p}{=} \left[\frac{1}{p^2} + \frac{1}{(1 - p)^2}\right](p - q)^2 + o\left((p - q^2)\right). \tag{52}$$

The expansion of the variance term is:

$$d_{\text{var}}(p, q) \underset{q \to p}{=} \left[\frac{1}{p^3} + \frac{1}{(1 - p)^3}\right](p - q)^2 + o\left((p - q^2)\right). \tag{53}$$

Matching the coefficients yields the tailored weight function:

$$w_\ell(q) = 2\left[\frac{1}{q^2} + \frac{1}{(1 - q)^2} + \frac{1}{q^3} + \frac{1}{(1 - q)^3}\right]. \tag{54}$$

We now recover the scoring rule by integrating the differential equations for strictly proper scores.

Using the relation $\frac{\partial}{\partial q}\ell(1, q) = -w_\ell(q)(1 - q)$:

$$\frac{\partial \ell(1, q)}{\partial q} = -\frac{2}{q^3} + \frac{2}{q} - \frac{2}{1 - q} - \frac{2}{(1 - q)^2}.$$

Integrating each term with respect to $q$:

$$\ell(1, q) = \frac{1}{q^2} + 2\log(q) + 2\log(1 - q) - \frac{2}{1 - q}. \tag{55}$$

Using the relation $\frac{\partial}{\partial q}\ell(0, q) = w_\ell(q)q$:

$$\frac{\partial \ell(0, q)}{\partial q} = \frac{2}{q} + \frac{2}{q^2} + \frac{2}{(1-q)^3} - \frac{2}{1-q}.$$

Integrating each term with respect to $q$:

$$\ell(0, q) = 2\log(q) - \frac{2}{q} + \frac{1}{(1-q)^2} + 2\log(1-q). \tag{56}$$

Combining the logarithmic terms into $\log(q(1-q))$ and assembling the final expression:

$$\ell(t, q) = t\left[\frac{1}{q^2} - \frac{2}{1-q} + 2\log\big(q(1-q)\big)\right] + (1-t)\left[\frac{1}{(1-q)^2} - \frac{2}{q} + 2\log\big(q(1-q)\big)\right]. \tag{57}$$

This matches Definition 3.7.

### F.3. Associated Canonical Mapping

We seek the canonical probability mapping $\sigma_\ell$ associated with the scoring rule $\ell$. By definition, : $(\sigma_\ell^{-1})'(p) = w_\ell(p)$. To find $\sigma_\ell$ we first integrate the weight function to find the relationship between $z$ and $p$. While the proof seem complicated, it just a pile of algebraic formulas that can be computed efficiently in python.

**1. Integral of the Weight Function** Recall $w_\ell(p) = \frac{2}{p^2} + \frac{2}{p^3} + \frac{2}{(1-p)^2} + \frac{2}{(1-p)^3}$. We integrate term by term, setting the constant of integration to zero to enforce symmetry $\sigma_\ell(0) = 0.5$:

$$\begin{aligned}
z &= \int \left(2p^{-2} + 2p^{-3} + 2(1-p)^{-2} + 2(1-p)^{-3}\right) dp \\
&= -\frac{2}{p} - \frac{1}{p^2} + \frac{2}{1-p} + \frac{1}{(1-p)^2} \\
&= \left(\frac{1}{(1-p)^2} - \frac{1}{p^2}\right) + 2\left(\frac{1}{1-p} - \frac{1}{p}\right).
\end{aligned} \tag{58}$$

**2. Change of Variable (The Quartic Equation)** Let $u = \frac{1}{p(1-p)}$. Since $p \in (0, 1)$, we have $u \geq 4$. We rewrite the terms of $z$ using $u$. Note that $\frac{1}{1-p} - \frac{1}{p} = \frac{2p-1}{p(1-p)} = u(2p-1)$. Similarly, the squared term is $u^2(2p-1)$. Thus:

$$z = (u^2 + 2u)(2p - 1). \tag{59}$$

Squaring both sides eliminates the dependence on the sign of $(2p - 1)$:

$$z^2 = (u^2 + 2u)^2(2p - 1)^2. \tag{60}$$

Substituting the identity $(2p - 1)^2 = 1 - 4p(1 - p) = 1 - \frac{4}{u} = \frac{u-4}{u}$:

$$z^2 = (u^2 + 2u)^2\left(\frac{u-4}{u}\right) = u(u+2)^2(u-4). \tag{61}$$

Expanding the polynomial on the right yields a depressed quartic equation in $u$:

$$\begin{aligned}
z^2 &= u(u^2 - 2u - 8)(u + 2) \\
z^2 &= u^4 - 12u^2 - 16u \\
0 &= u^4 - 12u^2 - 16u - z^2.
\end{aligned} \tag{62}$$

### 3. Analytic Solution via Ferrari's Method

We solve the quartic equation $u^4 - 12u^2 - 16u - z^2 = 0$ using Ferrari's method. The goal is to rewrite the equation as the equality of two perfect squares.

First, we move the linear and constant terms to the right side and add a parameter $y$ to "complete the square" on the left side for the term $(u^2 + y)^2$:

$$u^4 - 12u^2 = 16u + z^2$$
$$(u^2 + y)^2 = u^4 + 2yu^2 + y^2$$
$$= (12u^2 + 16u + z^2) + 2yu^2 + y^2$$
$$= (2y + 12)u^2 + 16u + (y^2 + z^2). \tag{63}$$

For the expression on the right-hand side of (63) to be a perfect square of the form $(ku + m)^2$, its discriminant with respect to $u$ must be zero. The quadratic is $Au^2 + Bu + C$ with $A = 2y + 12$, $B = 16$, and $C = y^2 + z^2$. The condition $\Delta = B^2 - 4AC = 0$ yields:

$$16^2 - 4(2y + 12)(y^2 + z^2) = 0$$
$$256 - 8(y + 6)(y^2 + z^2) = 0$$
$$32 - (y + 6)(y^2 + z^2) = 0. \tag{64}$$

Expanding this product gives the resolvent cubic equation in $y$:

$$32 - (y^3 + z^2 y + 6y^2 + 6z^2) = 0$$
$$y^3 + 6y^2 + z^2 y + (6z^2 - 32) = 0. \tag{65}$$

To solve this cubic, we eliminate the quadratic term $(y^2)$ using the substitution $y = 2v - 2$. We substitute this into (65):

$$(2v - 2)^3 + 6(2v - 2)^2 + z^2(2v - 2) + (6z^2 - 32) = 0. \tag{66}$$

Expanding the terms:

$$(8v^3 - 24v^2 + 24v - 8) + 6(4v^2 - 8v + 4) + (2z^2 v - 2z^2) + 6z^2 - 32 = 0$$
$$8v^3 + (-24v^2 + 24v^2) + (24v - 48v + 2z^2 v) + (-8 + 24 - 2z^2 + 6z^2 - 32) = 0$$
$$8v^3 + (2z^2 - 24)v + (4z^2 - 16) = 0.$$

Finally, dividing the entire equation by 8 yields the depressed cubic in $v$:

$$v^3 + \underbrace{\left(\frac{z^2}{4} - 3\right)}_{A} v + \underbrace{\left(\frac{z^2}{2} - 2\right)}_{B} = 0. \tag{67}$$

We define the discriminant of the depressed cubic as $\Delta = \left(\frac{B}{2}\right)^2 + \left(\frac{A}{3}\right)^3$. The solution for $v$ branches based on the sign of $\Delta$:

- **Case 1: $\Delta > 0$ (Hyperbolic Solution).**
  In this regime, the cubic has one real root and two complex conjugate roots. We use Cardano's formula to find the unique real root:
  $$v = \sqrt[3]{-\frac{B}{2} + \sqrt{\Delta}} + \sqrt[3]{-\frac{B}{2} - \sqrt{\Delta}}. \tag{68}$$

- **Case 2: $\Delta \leq 0$ (Trigonometric Solution).**
  In this regime we select the following root:
  $$v = 2\sqrt{-\frac{A}{3}} \cos\left(\frac{1}{3} \arccos\left(\frac{-B/2}{\sqrt{-(A/3)^3}}\right)\right). \tag{69}$$

Once $v$ is computed, we proceed to recover $u$ and subsequently $p$.

## 4. Recovering $u$ from the Perfect Square

The purpose of finding the root $v$ of the resolvent cubic is to force the right-hand side of our quartic equation to become a perfect square. This allows us to reduce the equation from degree 4 to degree 2.

Recall our equation from step 3:

$$(u^2 + y)^2 = (2y + 12)u^2 + 16u + (y^2 + z^2). \tag{70}$$

By setting $y = 2v - 2$, we ensured the discriminant of the quadratic on the right is zero. We can now write the coefficient of $u^2$ explicitly in terms of $v$:

$$2y + 12 = 2(2v - 2) + 12 = 4v + 8 = 4(v + 2). \tag{71}$$

Because the discriminant is zero, the entire quadratic on the right can be factored as a perfect square $(Mu + N)^2$, where $M = \sqrt{4(v + 2)} = 2\sqrt{v + 2}$. The constant term $N$ is determined by the linear coefficient $16 = 2MN$, which implies $N = \frac{8}{M} = \frac{4}{\sqrt{v+2}}$.

Thus, the original quartic equation becomes an equality of two perfect squares:

$$(u^2 + y)^2 = \left(2\sqrt{v + 2}\,u + \frac{4}{\sqrt{v + 2}}\right)^2. \tag{72}$$

Taking the square root of both sides allows us to solve a simple quadratic equation. We choose the positive branch to find the largest root for $u$:

$$u^2 + y = 2\sqrt{v + 2}\,u + \frac{4}{\sqrt{v + 2}}. \tag{73}$$

Rearranging terms to standard quadratic form $u^2 - Bu + C = 0$:

$$u^2 - \left(2\sqrt{v + 2}\right)u + \left(y - \frac{4}{\sqrt{v + 2}}\right) = 0. \tag{74}$$

Finally, solving for $u$ yields:

$$u = \sqrt{v + 2} + \sqrt{(v + 2) - \left(y - \frac{4}{\sqrt{v + 2}}\right)}. \tag{75}$$

Substituting $y = 2v - 2$ back into the expression gives the final formula:

$$u = \sqrt{v + 2} + \sqrt{4 - v + \frac{4}{\sqrt{v + 2}}}. \tag{76}$$

## 5. Recovering the Probability

We now invert the variable transformation to recover the probability $p$. Recall the definition:

$$u = \frac{1}{p(1 - p)}. \tag{77}$$

Rearranging this equation yields a standard quadratic equation in terms of $p$:

$$p(1 - p) = \frac{1}{u}$$
$$p^2 - p + \frac{1}{u} = 0. \tag{78}$$

Solving for $p$ using the quadratic formula:

$$p = \frac{1 \pm \sqrt{1 - \frac{4}{u}}}{2}. \tag{79}$$

Since the link function $g(p)$ is monotonic, the sign of the root is determined by the sign of the input $z$. The canonical link maps $p < 0.5$ to $z < 0$ and $p > 0.5$ to $z > 0$. Thus, we have:

$$p = \frac{1}{2}\left(1 + \text{sgn}(z)\sqrt{1 - \frac{4}{u}}\right). \tag{80}$$

This confirms that the inverse of the canonical link is analytically tractable and can be implemented efficiently without numerical optimization.

### F.4. Derivation of Bias-Only Scoring Rules

*Remark* F.1 (Bias-Variance Trade-off in Link Functions). We note that excluding the variance terms $(1/p^3)$ from the weight function simplifies the inversion to a quadratic equation, yielding a lighter "Bias-Only" loss. While computationally simpler, our experiments suggest that the full quartic link provides superior results.

Recall the bias component of the divergence from Theorem 3.5:

$$d_{\text{bias}}(p, q) = \left(\frac{p}{q} - 1\right)^2 + \left(\frac{1-p}{1-q} - 1\right)^2. \tag{81}$$

To apply curvature matching, we compute the second-order Taylor expansion around $q = p$:

$$d_{\text{bias}}(p, q) \underset{q \to p}{=} \left[\frac{1}{p^2} + \frac{1}{(1-p)^2}\right](p - q)^2 + o\left((p - q)^2\right). \tag{82}$$

Matching the coefficients yields the tailored weight function:

$$w_{\text{bias}}(q) = \frac{2}{q^2} + \frac{2}{(1-q)^2}. \tag{83}$$

We recover the scoring rule by integrating the differential equations for strictly proper scores.

Using the relation $\frac{\partial}{\partial q}\ell(1, q) = -w(q)(1 - q)$:

$$\frac{\partial \ell(1, q)}{\partial q} = -(1 - q)\left[\frac{2}{q^2} + \frac{2}{(1-q)^2}\right]$$
$$= -\frac{2}{q^2} + \frac{2}{q} - \frac{2}{1-q}.$$

Integrating with respect to $q$:

$$\ell(1, q) = \frac{2}{q} + 2\log(q) + 2\log(1 - q). \tag{84}$$

Using the relation $\frac{\partial}{\partial q}\ell(0, q) = w(q)q$:

$$\frac{\partial \ell(0, q)}{\partial q} = q\left[\frac{2}{q^2} + \frac{2}{(1-q)^2}\right]$$
$$= \frac{2}{q} + \frac{2}{(1-q)^2} - \frac{2}{1-q}.$$

Integrating with respect to $q$:

$$\ell(0, q) = 2\log(q) + \frac{2}{1-q} + 2\log(1 - q). \tag{85}$$

Combining terms yields the final Bias-Only scoring rule:

$$\ell_{\text{bias}}(y, q) = \frac{2y}{q} + \frac{2(1-y)}{1-q} + 2\log(q(1-q)). \tag{86}$$

### F.5. Associated Canonical Mapping

We seek the canonical link function $\sigma_\ell$ such that $(\sigma_\ell^{-1})'(p) = w_{\text{bias}}(p)$.

**1. Integral of the Weight Function** We integrate the weight function to find $z = \sigma_\ell^{-1}(p)$:

$$
\begin{aligned}
z &= \int \left( \frac{2}{p^2} + \frac{2}{(1-p)^2} \right) dp \\
&= -\frac{2}{p} + \frac{2}{1-p} \\
&= \frac{-2(1-p) + 2p}{p(1-p)} \\
&= \frac{2(2p-1)}{p(1-p)}.
\end{aligned}
\tag{87}
$$

**2. Inversion (The Quadratic Equation)** To find the inverse link $p = \sigma_\ell^{-1}(z)$, we solve Equation (87) for $p$. Rearranging the terms:

$$
\begin{aligned}
zp(1-p) &= 4p - 2 \\
zp - zp^2 &= 4p - 2 \\
zp^2 + (4-z)p - 2 &= 0.
\end{aligned}
\tag{88}
$$

This is a standard quadratic equation $Ap^2 + Bp + C = 0$ with $A = z$, $B = 4 - z$, and $C = -2$. Calculating the discriminant $\Delta$:

$$
\Delta = (4-z)^2 - 4(z)(-2) = 16 - 8z + z^2 + 8z = z^2 + 16.
\tag{89}
$$

Since $\Delta = z^2 + 16 > 0$ for all real $z$, real solutions always exist. Solving for $p$:

$$
p = \frac{-(4-z) \pm \sqrt{z^2 + 16}}{2z} = \frac{z - 4 \pm \sqrt{z^2 + 16}}{2z}.
\tag{90}
$$

We select the root that maps to the interval $(0, 1)$. For any $z \neq 0$, the valid root is given by the positive branch of the numerator relative to the sign of $z$. The unified formula is:

$$
p = \frac{z - 4 + \sqrt{z^2 + 16}}{2z}.
\tag{91}
$$

(Note: At $z = 0$, taking the limit yields $p = 0.5$).

