# OpenReview forum: "Tailoring Strictly Proper Scoring Rules for Downstream Tasks: An Application to Causal Inference"
_ICML.cc/2026/Conference — ICML 2026 regular_

### Official Review · Reviewer_rx3w · 2026-02-26

**Soundness:** 4
**Presentation:** 3
**Significance:** 3
**Originality:** 4
**Overall Recommendation:** 5
**Confidence:** 3

**Summary:**

(EDIT: final score edited due to review)

A classical issue in inverse probability weights estimators is that propensity models are fitted using canonical classification losses such as binary cross-entropy that do not account for the propensity model inversion, potentially leading to large errors. Generally, it is common to fit probabilistic models using general methods and then use them in downstream tasks, leading to suboptimal performance as the general methods are not adapted to downstream tasks. This paper focuses on binary classification models and proposes to choose proper scoring rules that are adapted to the task at hand. A strategy to recover the scoring rule is presented for general tasks, and applied to average treatment effect (ATE) estimation in causal inference; this gives a new loss to fit the propensity score. The resulting loss is found to have competitive performance against classical inverse propensity weight estimators, including those relying on covariate balance or balancing weights.

**Compliance With Llm Reviewing Policy:**

Affirmed.

**Final Justification:**

The rebuttal addressed all my concerns, the theory and experiments look great. I would generally rate the impact as "high" rather than "exceptional".

**Key Questions For Authors:**

Minor clarity question but in Proposition 3.3, shouldn't $w_{\text{task}}$ be used instead of $w_{\ell}$ to better underline the fact that we are recovering $\ell$ from the task?

Remark 3.9 suggests that, in general case, a root-finding routine is applied and a manual gradient is specified for the backward pass. But from my understanding, for the IPW estimator, Equation (17) is fully closed-form, with the largest root given in Equation (62)? And the specification of the MLP (model 2 in Appendix B.2) does not explain whether the default autograd gradient or a custom one is applied, can you specify this?

In Figure 1, for which metric is the Mean Rank relative to? The absolute bias, the RMSE or the MAE? Or all of them (with the three ranks averaged)? And how are overlaid scatter points computed from the individual runs? The median of the relevant runs?

l.423-428: "A strength of our approach is its modularity. Unlike methods that require specialized architectures or complex optimization routines, our framework provides a plug-and-play loss and probability mapping pair. This acts as a drop-in replacement for the standard log-loss, allowing practitioners to leverage the tailored objective across any model." I do not understand this. The method does rely on specific MLE or XGBoost implementations, notably using custom derivatives for the latter?

I am ready to increase my score if the above points are addressed.

**Limitations:**

Limitations are not indicated, to the best of my understanding. Those could include points such as the potential difficulty of finding a closed-form scoring rule in the general case, or the computational burden of the root-finding routine in the same general case.

**Strengths And Weaknesses:**

**Soundness** : the submission appears technically sound. I have checked the proofs and those all look correct. Experiments are thorough and demonstrate the claims of the papers. See questions below however. A discussion of lmitations also seems to be missing.

**Presentation** : the paper is generally clearly written and easy to follow. The work positions itself in the context of prior literature. Some minor catches are :
(a) It is never explicitly defined what the entropy $H_\ell$ is exactly, even if the information about its second derivative can help the reader understand that it is $\mathbb{E}_{y \sim p}[\ell(y,p)]$ (please correct me if I am wrong).
(b) The variable $T$ is used in Proposition 3.4, in contrast to $y$ used before.

**Signifiance** : the method is significant as it introduces a well-justified method to circumvent the ubuquitous difficulty in fitting propensity scores that are suited to inverse probability weighting. It also introduces a more general framework to design proper scoring rules for any dowstream tasks, even if I am not knowledgable on whether such a framework is actually novel or has been proposed before.

**Originality** : the paper introduces a novel way to choose a proper scoring rule and the associated link function for probabilistic model fitting, notably drawing on past theory on proper scoring rules. As a result, for treatment effect estimation, it presents a dramatical improvement over the previous use of proper scoring rules in Zhao (2019) choosing a different scoring rule (and link function) and justifying it through the task's mean squared error. Derivations to obtain the scoring rule are non-trivial, and the result on the appropriate link function is elegant. Novelty is generally well-justified.

---

> ### Author Rebuttal · Authors · 2026-03-30
>
> We thank the reviewer for their thorough review and for recognizing the theoretical soundness of our derivations. We have addressed all of your points below and will update the paper accordingly.
>
> **On the reviewer presentation remarks:**
>
> Entropy is, in fact, the quantity the reviewer refers to; we will explicitly incorporate this definition in the Related work part. We will also harmonize notation between $T$ and $y$, reformulate Prop 3.3, and use $w_\text{task}$ instead of $w_\ell$.
>
> **On remark 3.9, lines 423-428, and Autograd use:**
>
> We thank the reviewer for this question and for giving us the opportunity to make things clearer. We acknowledge that our text may have been slightly confusing by not separating our practical implementation from the actual requirements of the method.
>
> The reviewer is correct that the forward pass is fully closed-form via Equation 62. To clarify the backward pass, practitioners actually have multiple paths to implement our method depending on their chosen architecture:
>
> For MLP/GLMs/Deep Learning architectures they can
>
> 1. Rely on standard autograd: Because the root-finding routine consists of standard differentiable operations, practitioners can simply use the closed-form forward pass and let standard automatic differentiation handle the backward pass. In this sense, it acts as a "drop-in replacement" for log-loss that requires zero custom gradient code, though it introduces a computational overhead when backpropagating through the mapping.
>
> 2. Implement a custom backward pass: For our specific MLP implementation, we chose to implement a custom backward pass for computational efficiency. Because our tailored loss and probability mapping form a canonical pair, we theoretically **know** that the analytical gradient with respect to the logits simplifies to $p - T$. By doing so, we completely bypass the need to backpropagate through multiple steps of the quartic equation solver.
>
> For gradient boosting:
>
> 1. Native Custom Objectives: For frameworks like XGBoost, the concept of differentiating through a computational graph does not apply. Instead, libraries like sklearn requires the analytical gradient and Hessian of the loss with respect to the logits. Thanks to our canonical link, these derivatives simplify to $p - T$ and $\frac{1}{w_\ell(p)}$ . By supplying these directly, we do not need to differentiate through the complex probability mapping.
>
> We will update the manuscript and Appendix B.2 to explicitly outline these implementation paths and highlight our specific implementation choice.
>
> **On Mean Rank Metric:**
>
> The mean rank is computed per-simulation based on the squared error relative to the ground truth ATE. (Because we are evaluating at a per-simulation level, ranking by squared error or absolute error yields the exact same rankings). These ranks are first averaged within each dataset, and then those averages are macro-averaged across all three datasets to produce the final score.
>
> **Overlaid Scatter Points:**
> The overlaid scatter points represent the average performance, not the median. Specifically, a method’s error is first normalized by dividing it by the median error of all methods within that specific simulation. The scatter point plots the mean of these normalized errors across all simulations. Therefore, a mean value $< 1.0$ means that, on average, the method performs better than the median method, while a value $> 1.0$ means it performs worse.
>
> **On Limitations Section**. We thank the reviewer for pointing this out. We completely agree and will add a dedicated Limitations paragraph. As the reviewer rightly notes, we will explicitly mention that while the framework is general, finding a closed-form analytic solution for the canonical link function $\sigma_l$ requires solving polynomial equations, which may not be analytically tractable for more complex downstream error bounds. Furthermore, even when tractable (like our quartic solver), it introduces a computational overhead during the forward pass compared to a standard Sigmoid activation.
>
> Once again, we thank the reviewer for their constructive feedback, which will improve the current version of our paper.

---

> > ### Author Rebuttal · Reviewer_rx3w · 2026-04-02
> >
> > Many thanks! All my concerns are resolved except these two points:
> >
> > (a) By the "modularity" of your method, what do you mean then? That it just replaces log-loss, and can do so in any method implementing logistic regression?
> >
> > (b) I find it odd that the overlaid scatter points represent the average, given that they are superposed on boxplots representing quantiles such as the median and the 25th/75th percentiles. How does the figure change if you replace averages with medians?

---

> > > ### Author Response · Authors · 2026-04-02
> > >
> > > **On modularity**
> > >
> > > Yes, exactly. By modularity, we mean that our framework acts as a direct replacement for the standard log-loss + sigmoid pair, and can be used in any gradient-based model (e.g., MLPs or gradient boosting) without modifying the architecture.
> > >
> > > **On mean vs. median scatter points**
> > >
> > > We thank the reviewer for this insightful remark and agree that medians are more standard when paired with boxplots. Our initial choice to display means was motivated by the specific failure modes of IPW estimators. In particular, baseline methods (e.g., logistic regression) can produce extreme propensity scores near 0 or 1, leading to bias and variance explosions. Because the mean is sensitive to such extreme values, it provides a diagnostic of robustness to these failure cases.
> > >
> > > That said, we agree that this choice may appear inconsistent with the boxplot visualization. To address this, we evaluated the impact of replacing the overlaid means with medians. The underlying distributions (boxplots and rankings) remain unchanged. Using medians for the scatter points actually slightly strengthens our empirical findings. As shown in the table below, our method improves over baselines under both metrics. Notably, the relative improvement is even larger under the median: across datasets and for the IPW estimator, our custom loss is on average (across datasets) about 2× better than the median baseline in terms of mean, and about 4× better in terms of median. We will clarify this point in the final version and consider updating the visualization for better consistency.
> > >
> > >
> > > **Table : Relative error vs. median baseline (Mean vs. Median aggregation)**
> > > | Method | IHDP (Mean) | IHDP (Median) | Jobs (Mean) | Jobs (Median) | Kang-Schafer (Mean) | Kang-Schafer (Median) |
> > > | :--- | :---: | :---: | :---: | :---: | :---: | :---: |
> > > | SBW | 8.30x | 5.96x | 0.76x | 0.45x | 1.78x | 1.40x |
> > > | Logistic | 0.70x | 0.62x | 3.00x | 1.82x | 8.03x | 1.02x |
> > > | CBPS | 2.39x | 1.94x | 1.35x | 1.53x | 0.78x | 0.50x |
> > > | CBSR | 55.09x | 42.84x | 10.22x | 6.67x | 54.03x | 42.80x |
> > > | Clipping | 0.71x | 0.69x | 3.00x | 1.80x | 0.96x | 0.94x |
> > > | Trimming | 1.10x | 1.15x | 0.18x | 0.10x | 1.05x | 0.91x |
> > > | EB | 0.52x | 0.28x | 0.35x | 0.33x | 1.29x | 1.01x |
> > > | Custom (Ours) | 0.55x | 0.34x | 0.24x | 0.08x | 0.46x | 0.34x |

---

### Official Review · Reviewer_8mHD · 2026-03-12

**Soundness:** 3
**Presentation:** 3
**Significance:** 3
**Originality:** 3
**Overall Recommendation:** 4
**Confidence:** 4

**Summary:**

The paper proposes a new method for estimating propensity scores that are better suited for causal effect estimation, particularly for IPW estimators of the ATE. The key idea is that standard training objectives for propensity score models—such as logistic regression with log-loss—are designed for accurate probability prediction but are not aligned with the downstream causal estimation task, where errors in propensity scores near 0 or 1 can greatly inflate bias and variance. The authors derive a task-aware loss function by analyzing the mean squared error of the IPW estimator and constructing a tailored strictly proper scoring rule whose curvature penalizes errors in extreme propensity regions more heavily. They also derive a corresponding canonical link function to ensure stable optimization when training the model. The resulting approach trains propensity score models using this customized loss and then uses the estimated scores in standard causal estimators such as IPW or AIPW, showing improved empirical performance on several benchmark datasets.

**Compliance With Llm Reviewing Policy:**

Affirmed.

**Key Questions For Authors:**

1. It is not entirely clear how sensitive the performance is to deviations from the assumptions used in this derivation (e.g., bounded outcomes or well-behaved propensity scores). Could the authors clarify how robust the method is when these assumptions are violated in practice?
2. The method focuses on improving propensity score estimation for downstream IPW-based estimators. Have the authors evaluated how the tailored loss performs when the estimated propensity scores are used with other causal estimators, such as doubly robust or targeted learning approaches? It would be helpful to understand whether the benefits extend beyond IPW.
3. Since the tailored scoring rule heavily penalizes propensity scores near 0 or 1, how does the method behave in scenarios with limited overlap or extreme treatment assignment probabilities? Are there situations where the proposed loss could introduce instability or bias?

**Limitations:**

See the weakness.

**Strengths And Weaknesses:**

Strengths: The paper addresses an important issue in causal inference: the mismatch between standard probability prediction objectives and the downstream goal of accurate causal effect estimation. By designing a task-aware loss for propensity score estimation based on the error structure of IPW estimators, the work provides an interesting perspective on aligning model training with causal estimation objectives.
Weakness: The methodological novelty appears somewhat limited, as the main contribution is a reformulation of the training loss rather than a fundamentally new causal estimation framework. The theoretical analysis mainly motivates the proposed loss but does not provide strong guarantees about improvements in causal effect estimation beyond the derived upper bounds.

---

> ### Author Rebuttal · Authors · 2026-03-30
>
> We thank the reviewer for the feedback and for recognizing the relevance of our work. We address your specific questions below.
>
> **On novelty and theoretical guarantees**
>
> We view the derivation of a task-specific strictly proper scoring rule and its exact canonical link as a novel methodological contribution. Rather than a simple loss reformulation, it creates a direct mathematical bridge between probability learning and downstream estimation error. Regarding theoretical guarantees, we agree that our analysis establishes an upper bound to motivate the loss, rather than providing strict guarantees for the final estimand, which we instead validate through our extensive empirical results.
>
> **Sensitivity to Assumptions**
>
> 1. Bounded outcomes: To the best of our knowledge, this is a standard regularity condition in causal inference required to ensure finite variance. In practice, most continuous outcomes are bounded. We believe that analyzing unbounded outcomes scenarios falls outside the scope of this study.
>
> 2. Well-behaved propensity scores: Robustness against the violation of this assumption is one of the main contributions of our tailored loss. Standard log-loss estimators suffer from bias and variance inflation when propensity scores approach 0 or 1. Our method, by matching the curvature of the downstream error, creates a loss whose local sensitivity is the same as the MSE of the ATE estimator. To test, in practice, this robustness we use the Kang \& Schafer benchmark and the ACIC 2017 "High selection Strength" datasets regimes (in which propensity scores are not well-behaved *i.e* poor overlap occurs). Our method consistently dominates in these scenarios.
>
> **Extension to Doubly Robust Estimators**
>
> Our method does indeed extend beyond standard IPW. In fact, we already evaluate our method using a Doubly Robust estimator (AIPW) in our experiments. As shown in Table 2 and Figure 4, the propensity scores trained with our tailored loss improves the downstream performance of the AIPW estimator (except for the very right plot of Figure 4). Furthermore, we provide the formal theoretical proofs in Appendix D.3 demonstrating that the theoretical upper bound of the AIPW estimator is governed by the exact same propensity divergence ($d_{task}$) as IPW. Regarding TMLE we stated in the last paragraph of the conclusion that we believe our framework could be extended to other estimators, as AIPW and TMLE are known to be asymptotically equivalent.
>
> **Instability, Bias, and Limited Overlap Instability:**
>
> - Heavily penalizing extreme probabilities with $O(1/p^3)$ weights would intuitively lead to gradient explosion. However, we explicitly solve this mathematically in Section 3.2.4 by deriving the canonical link function (Definition 3.8). By pairing our tailored loss with this specific activation function, the gradients simplify to a stable $p - T$. We empirically show this stability in Appendix A.1 (Figure 5).
> - Bias in limited overlap: The canonical link effectively "stretches" the probability space (Figure 6), making it harder for the model to predict extreme boundary values. We did not empirically nor theoretically study if this could lead to a structural bias but ultimately, as shown in the "Bias" panel of Figure 2, the downstream ATE bias of our estimators remains the closest to the true target.
>
> We warmly thank the reviewer again for their insightful comments.

---

> > ### Author Rebuttal · Reviewer_8mHD · 2026-04-05
> >
> > I will keep my score.

---

### Official Review · Reviewer_5ULc · 2026-03-12

**Soundness:** 3
**Presentation:** 1
**Significance:** 3
**Originality:** 3
**Overall Recommendation:** 4
**Confidence:** 2

**Summary:**

Many modern causal inference methods involve plugging in predictions from flexible machine learning models. Traditionally, training the predictive model and constructing an estimator are treated as separate tasks. But this can lead to problems, even when the ML model is trained using a proper scoring rule (which is minimized at the truth). For example, perhaps the model often outputs predicted propensity scores very close to 1 or 0, leading IPW estimator variances to blow up. This paper considers the problem of designing proper scoring rules so that the resulting predictors behave well with respect to the downstream inference task.

Their main theory result is a method for deriving training loss functions that are (1) convex and (2) penalize errors in proportion to their impact on the downstream task.  They instantiate the result to derive a proper loss tailored to minimizing the MSE of an IPW estimator for the ATE.

Empirically, they evaluate their work on a standard set of semisynthetic datasets (ACIC 2017) where individual treatment effects are observed. Their method often performs better than alternatives.

**Compliance With Llm Reviewing Policy:**

Affirmed.

**Final Justification:**

I remain positive on the paper. I would have rated it higher if not for exposition/presentation issues that make it harder to evaluate the paper.

**Key Questions For Authors:**

N/A

**Limitations:**

I felt there was inadequate discussion of the following: In practice, it may still make sense to use generic loss functions. For example, practically, researchers may not want to implement a new training procedure that optimizes some non-standard loss. Moreover, libraries may be optimized for generic losses so that the computational or other practical costs of running training for a specialized loss are not worth the benefits.

**Strengths And Weaknesses:**

Strengths:

1. Developing a tighter relationship between methods for estimating nuisance parameters (e.g., propensity score predictors) and the target estimation task may yield practical improvements in the quality of semiparametric statistical inference. This is a good area to work on since my perception is that most nuisance parameter estimation happens via generic ML training (e.g., sklearn) and it is worth exploring whether something more tailored works better.
2. From a theoretical perspective, it is also interesting to analyze the properties of proper scoring rules that make them favorable or unfavorable to downstream statistical inference tasks. This paper takes a first step towards that goal.
3. I really appreciated the fact that this work analyzes a very concrete and common task: estimating an ATE using an IPW estimator. The crisp instantiation of their general theory in a simple and common example is useful for both general intuition and because the IPW example is of interest on its own.

Weaknesses:

1. At an expositional level, I think this paper could be improved. Text around the results was frequently missing context about why the results were being presented and what role they play in the overall analysis. In particular, I would like the authors to clarify the purpose of the canonical link function. Is it only relevant to the final activation function of a neural network? Or perhaps only GLMs? Or all gradient-based methods? The authors should clarify what ML methods are consistent with this framework and which are ruled out. I am marking low-confidence in my evaluation in part because I found it hard to parse the results and still do not understand the role of the canonical link.
2. The empirical results were difficult to interpret. This is in part because of their choice of visualization. Determining whether a particular dataset and estimator is outperformed by another method involves matching colors/shapes (often overlapping) across the different rows of the plots. It would be much better to seperate these out into a separate plot for IPW, Hajek and AIPW (and perhaps even by dataset) so that there aren’t 9 different points that each need to be individually traced through the plots. It would be fine to just cover one estimator in the body of the paper and present the other results in the appendix. Similar comments apply to Figure 3, where many different data generating processes are analyzed.

---

> ### Author Rebuttal · Authors · 2026-03-30
>
> We thank the reviewer for their review and feedback. We hope our answers will help clarify some key concepts of the manuscript.
>
> **On the role and the scope of the canonical link function**. As requested by the reviewer, we clarify below its role.
>
> In standard binary classification, practitioners typically pair the Binary Cross-Entropy with the Sigmoid activation function. The Sigmoid is the "canonical link" (actually the inverse canonical link, but it is a wording convention) for BCE because the gradient with respect to logits gives the simple quantity $p - T$ (where $p$ is the prediction and $T$ is the true label).
>
> Because we designed a new loss function with massive penalties near 0 and 1, pairing it with a standard Sigmoid would cause the gradients to explode. To fix this, we derived the new canonical link function (Definition 3.8) specifically for our tailored loss. By replacing the Sigmoid with this new activation function at the final layer of the model, the gradients with respect to logits simplify, again to $p - T$.
>
> We will update Section 3.2.4 to clarify the role of the canonical link function, as well as the related work section to add the example of the Sigmoid for BCE.
>
> **On Presentation and Narrative Flow:** In addition to clarifying the canonical link, we will revise the transition paragraphs throughout Section 3 to better contextualize our theoretical results before introducing them.
>
> **On the Scope:** This framework applies broadly to any gradient-based model that outputs a logit which is then mapped to a probability. This includes Neural Networks, XGBoost/Gradient Boosted Trees, GLMs, and so on.
>
> We will add a paragraph before the Experiments section that explicitly provides this context and clearly lists the compatible architectures.
>
> **On the visualization interpretation**.
>
> Our initial goal was to provide a holistic perspective allowing for the comparison of methods across the 'dataset' and 'estimator' dimensions. However, we agree that this sacrificed simplicity and resulted in dense plots. Following your recommendation, we will replace the aggregated Figures 2 and 3 in the main text with separated plots broken down by estimator (IPW, Hajek, AIPW). We will move the aggregate plots to the Appendix for completeness.
>
> **Addressing Practical Considerations**
>
> We thank the reviewer for highlighting these practical considerations. We address these concerns below and will update the manuscript to make these trade-offs clear.
>
> - **No "New Training Procedure" Required.** First, we would like to clarify that our framework does not require a new training procedure or any custom optimization procedures. The training loop, model architecture, and optimization algorithms remain exactly the same: practitioners simply swap in our tailored loss and its corresponding canonical probability mapping as a plug-and-play drop-in replacement for the standard BCE.
>
> - **Optimized Backward Pass.** We completely agree that computing our custom objective introduces a computational overhead, as generic losses benefit from highly optimized, low-level library implementations. For MLP implementations, we mitigate this by restricting the quartic equation solving strictly to the forward pass. Because our loss and probability mapping form a canonical pair, the analytical gradient simplifies to exactly $p - T$. By providing this custom backward pass (as noted in Remark 3.9), we entirely bypass the need to backpropagate through the root-finding procedure. Therefore, the backward pass, which is typically the computational bottleneck, remains exactly as cheap as standard log-loss. Similarly, for XGBoost, we explicitly supply the exact analytical gradient and Hessian.
>
> - **The "Bias-Only" Alternative**. If computational cost remains a strict constraint, practitioners can use our "Bias-Only" scoring rule. By dropping the variance penalty from the weight function, the forward-pass probability mapping simplifies to a quadratic equation. As demonstrated in our ablation study, this lighter alternative outperforms the standard log-loss in difficult regimes while being computationally cheaper.
>
> - **Manuscript Updates: Limitations Section.** We agree that this trade-off between downstream estimation accuracy and computational cost should be explicitly stated. We will therefore add a Limitations section. Within this section, we will include concrete timing metrics for the standard log-loss, the bias-only loss, and the full-MSE objective.
>
> We warmly thank the reviewer again for their insightful comments.

---

> > ### Author Rebuttal · Reviewer_5ULc · 2026-04-03
> >
> > Thanks to the authors for their clarifications. I remain positive about this paper

---

### Official Review · Reviewer_chiV · 2026-03-13

**Soundness:** 3
**Presentation:** 3
**Significance:** 3
**Originality:** 3
**Overall Recommendation:** 4
**Confidence:** 3

**Summary:**

This paper proposes a general framework for designing task-specific strictly proper scoring rules by matching the local curvature of the downstream error functional. Specializing to ATE estimation via IPW, the authors derive (i) an explicit strictly proper loss whose curvature matches an upper bound on the MSE of the IPW estimator and (ii) its associated canonical probability mapping, ensuring convexity in the logits and stable optimization. Experiments on semi-synthetic causal benchmarks and ACIC 2017 show strong improvements over log-loss, with the clearest gains in high-selection-strength regimes.

**Compliance With Llm Reviewing Policy:**

Affirmed.

**Final Justification:**

My questions have been answered. I will keep my positive overall recommendation.

**Key Questions For Authors:**

1. Can Theorem 3.5 be restated explicitly under the sample-splitting or cross-fitting assumptions actually used in the experiments, and were both the propensity and outcome models cross-fit in all AIPW experiments?
2. Can the paper more fully substantiate the extension beyond IPW, especially by providing the missing Hajek proof details, constants, and precise assumptions, and by clarifying how the Appendix D.3 AIPW derivation supports the claim in Remark 3.6?
3. Can the paper calibrate some broader-sounding phrases, such as "minimizes the upper bound" and "theoretically optimal objective," so they more closely match the local scope of the curvature-matching argument?

This topic is somewhat outside my core area of expertise, and I would be open to revising my scores after reading the authors' clarification.

**Limitations:**

While the paper states the causal assumptions and describes cross-fitting in the experimental setup, the limitations discussion is too generic. It would be helpful to discuss more concretely the role of out-of-sample nuisance estimation in the theory, the fact that the curvature argument is local, and the practical risks of applying the method when the standard causal assumptions or overlap conditions fail.

**Strengths And Weaknesses:**

Strengths

- The paper presents a principled general framework for constructing downstream-aware strictly proper scoring rules via local curvature matching, together with a concrete instantiation for propensity estimation and ATE estimation via IPW.
- The empirical section shows broad gains across classic semi-synthetic benchmarks and ACIC 2017, including IPW, Hajek, and AIPW evaluations, with the clearest improvements in weak-overlap or high-selection-strength regimes.
- The canonical-link construction is technically interesting, and retaining the canonical link helps preserve convexity and stable optimization.
- The experimental evaluation spans classic semi-synthetic benchmarks and ACIC 2017 with multiple estimators, backbones, cross-fitting, and useful ablations.
- The approach targets an important problem in propensity-score estimation under weak overlap and seems practically useful as a drop-in objective.

Weaknesses

- Theorem 3.5 would benefit from clarifying whether the estimated propensity score is treated as fixed or out-of-sample, since the proof writes expectations as if the estimated propensity score were treated as fixed, while the experiments use cross-fitting.
- The extension beyond IPW could be substantiated more fully: the paper includes an AIPW derivation in Appendix D.3, but the Hajek extension remains informal, and the connection between Appendix D.3 and the broader claim in Remark 3.6 could be clarified.
- The paper already makes clear that the curvature-matching argument is local, but some stronger phrases, such as the discussion around "minimizes the upper bound" and Remark 3.6's "theoretically optimal objective," could be calibrated to better match that local scope.
- Several baselines rely on custom or adapted implementations, and although Appendix B provides useful details, the paper could clarify more precisely how closely these implementations match standard reference versions; it is also unclear whether AIPW outcome models were cross-fit symmetrically with the propensity models.

---

> ### Author Rebuttal · Authors · 2026-03-30
>
> We thank the reviewer for their thorough review of our work and their highly relevant remarks and questions. We hope the following clarifications address your concerns.
>
> **About Th. 3.5 and its underlying assumptions**.
>
> We thank the reviewer for this point. The derivations in Th. 3.5 are based on the assumption that the estimated propensity score $\hat{e}(X)$ is an out-of-sample estimate, allowing it to be treated as a fixed function. We agree that this non-trivial assumption was left implicit, and we detail below why it is actually met in practice **thanks to** cross-fitting.
>
> Cross-fitting consists of training $K$ separate models (each trained on $K-1$ folds). When evaluating $\hat{e}(X_i)$, we strictly use the model for which unit $i$ was not included in the training set. By doing so, we ensure that $(X_i, T_i, Y_i)$ is strictly independent of the fitted model, therefore allowing $\hat{e}$ to be treated as fixed. We realize that this should have been stated more clearly and will reformulate Th. 3.5 to explicitly account for the cross-fitting assumptions. We will also add a detailed explanatory paragraph in the Appendix.
>
> **Regarding the AIPW experiments**
>
> Yes, both the propensity models and the outcome models were cross-fitted symmetrically.
>
> **Clarifying Remark 3.6 and the Hajek Estimator**
>
> We take the opportunity of the reviewer remark to clarify Remark 3.6 and will include the full Hajek proof in the revised Appendix.
>
> The core claim of Remark 3.6 is that the theoretical MSE upper bounds for the standard IPW, Hajek, and AIPW estimators all share the same mathematical structure: the estimation error for all three estimators is driven by the same task-specific divergence $d_{\text{task}}(e(X), \hat{e}(X))$. While the scaling constants ($C_0$, $C_1$) differ across the estimators, the underlying divergence metric remains identical. This is how the connection between Appendix D.3 and the broader claim in Remark 3.6 is made.
>
> **Clarification of the precise Assumptions**
>
> The assumptions are all stated in the manuscript, but we agree they could have been centralized.
> We recall them all here:
> - Standard causal inference assumptions: Unconfoundedness, SUTVA, and positivity.
> - Bounded outcome variance: The conditional second moments of the potential outcomes are bounded (stated in Th. 3.5).
> - Cross-Fitting: The propensity model $\hat{e}(X)$ is fixed (cf. answer above).
>
> **Overview of the Hajek Proof** (Will be added in Appendix D.3). We give below an overview of the missing proof.
>
> Because the Hajek estimator is a ratio of two empirical sums, computing an exact finite-sample MSE is not feasible. The proof therefore derives an asymptotic bound using a first-order Taylor expansion around the true expectations to linearize the error of the ratio. This linearization reveals that the asymptotic error of the Hajek estimator shares the exact same mathematical structure as the standard IPW estimator. The only difference is that it applies IPW formula to a centered outcome instead of the raw outcome. Because this linearized error structurally matches standard IPW, squaring it to bound the MSE follows the exact same mathematical steps. Therefore, the asymptotic MSE bound is driven by the same $d_{\text{task}}(e(X), \hat{e}(X))$ divergence derived for standard IPW.
>
> **On the Tailored Loss and Bounds**.
>
> We thank the reviewer for this fair critique. As the reviewer correctly understood, while the theoretical upper bound on the MSE (Th. 3.5) is a globally valid inequality, our tailored loss is constructed to approximate this bound locally via curvature matching. Throughout the manuscript, we will reformulate the mentioned sentences to avoid giving the impression of overstating the global scope of the result.
>
> **On Baseline Implementations**
>
> As stated in the Appendix, for CBPS, we used a custom implementation that optimizes a joint objective (negative log-likelihood and a covariate balance penalty), essentially matching the method proposed by Shang et al. (2025). For CBSR, we directly implemented the theoretical loss function derived by Zhao (2019), applying stronger L2 regularization to prevent gradient explosion during optimization with L-BFGS-B. Finally, for SBW and EB, as, to the best of our knowledge, no gold standard Python implementations exist, we simply translated the R implementations into Python.
>
> **On Limitations**
>
> We will add a Limitations section synthesizing the points raised above regarding out-of-sample estimation and local curvature-matching. We will also clarify the practical risks: while our method mitigates weak overlap, strict overlap failures will still cause structural bias. Finally, we believe that the study of other strict causal assumption violations (e.g., unconfoundedness) falls outside the scope of our research, but we will explicitly mention this in the limitations.
>
> Once again, we thank the reviewer for their constructive feedback, which will improve the current version of our paper.

---

> > ### Author Rebuttal · Reviewer_chiV · 2026-04-01
> >
> > Thank you for the detailed rebuttal. Your response addresses my main concerns. I also appreciate the added detail on baseline implementations and the plan to add explanatory paragraphs/sections. I am keeping my overall positive recommendation.

---

### Decision · Program_Chairs · 2026-04-30

**Decision:**

Accept (regular)

**Comment:**

This submission proposes a framework for tailoring strictly proper scoring rules to downstream tasks by matching the local curvature of the downstream objective, and applies the framework to causal inference via propensity score estimation for IPW-type estimators. The manuscript is technically solid, the central idea is clean, and the causal application is meaningful.

The reviewers raised reasonable questions about scope, assumptions, phrasing, and empirical presentation, but these concerns were largely addressed in rebuttal. Overall, I believe the paper makes a solid contribution that fits ICML well. I therefore recommend Accept.